# Deep Brain Stimulation Frequency—A Divining Rod for New and Novel Concepts of Nervous System Function and Therapy

**DOI:** 10.3390/brainsci6030034

**Published:** 2016-08-17

**Authors:** Erwin B. Montgomery, Huang He

**Affiliations:** Greenville Neuromodulation Center, 179 Main St., Greenville, PA 16125, USA; Hehuang@grnneuromod.com

**Keywords:** Deep Brain Stimulation (DBS), stimulation frequency, discrete nonlinear oscillators, stochastic resonance, basal ganglia–thalamic–cortical system of oscillators, Principle of Causational Synonymy, Principle of Informational Synonymy

## Abstract

The efficacy of Deep Brain Stimulation (DBS) for an expanding array of neurological and psychiatric disorders demonstrates directly that DBS affects the basic electroneurophysiological mechanisms of the brain. The increasing array of active electrode configurations, stimulation currents, pulse widths, frequencies, and pulse patterns provides valuable tools to probe electroneurophysiological mechanisms. The extension of basic electroneurophysiological and anatomical concepts using sophisticated computational modeling and simulation has provided relatively straightforward explanations of all the DBS parameters except frequency. This article summarizes current thought about frequency and relevant observations. Current methodological and conceptual errors are critically examined in the hope that future work will not replicate these errors. One possible alternative theory is presented to provide a contrast to many current theories. DBS, conceptually, is a noisy discrete oscillator interacting with the basal ganglia–thalamic–cortical system of multiple re-entrant, discrete oscillators. Implications for positive and negative resonance, stochastic resonance and coherence, noisy synchronization, and holographic memory (related to movement generation) are presented. The time course of DBS neuronal responses demonstrates evolution of the DBS response consistent with the dynamics of re-entrant mechanisms. Finally, computational modeling demonstrates identical dynamics as seen by neuronal activities recorded from human and nonhuman primates, illustrating the differences of discrete from continuous harmonic oscillators and the power of conceptualizing the nervous system as composed on interacting discrete nonlinear oscillators.

## 1. The Conundrum of Deep Brain Stimulation (DBS) Frequency

The effects of specific electrode configurations, the set of active cathodes and anodes, can be understood to relate to the regional anatomy and the volume of tissue activation. While the effects of stimulation parameters, such as stimulation current (voltage) and pulse width, can be understood as related to activation of neuronal axonal elements mediated by the presence or absence of myelin, axonal diameter, and chronaxie, how DBS frequency controls the response is much more problematic [1]. From the perspective of information transfer between neurons, such as those activated directly by the DBS pulse and the subsequent postsynaptic neurons, DBS frequency could have an effect on temporal summation. Such temporal summation would be important in propagating the DBS effect through a long sequence of interactions, ultimately affecting the orchestration of motor unit activities that mediate the clinical motor responses to DBS. However, the frequencies used typically in DBS are not very effective at temporal summation [2], and thus the mechanisms by which DBS frequencies affect motor control likely are through some other mechanism(s).

The neurophysiology of the DBS frequency effect is an enigma and likely will require novel attempts to explicate. This article takes on this challenge, proceeding from the hypothesis that DBS can be considered a nonlinear discrete noise oscillator that is interjected into the nonlinear polysynaptic re-entrant discrete oscillators that comprise the basal ganglia–thalamic–cortical system. Whether this will illuminate the issue of DBS frequency effects requires the test of time. However, the theory requires the introduction of novel or, at the very least, unfamiliar concepts—hence the review-like nature of the article; these will be in addition to novel observations, hence the presentation of experiments. The format provides a significant advantage in the opportunity to make a full proposal with the mutual reinforcement or consilience of the many parts not typically available in most journals.

The first effort is to demonstrate just how complex the effect of varying DBS frequencies is, in this case the effect of various DBS frequencies in the vicinity of the subthalamic nucleus on bradykinesia in patients with Parkinson’s disease, and to account for the failure of such complexity to be appreciated previously. There is considerable experience that demonstrates benefits from low-frequency DBS, for example of the STN for gait disturbances and for dystonia. Further, there may be differences in the frequency-dependent responses for different symptoms and signs of Parkinson’s disease. The failure to appreciate this complexity, although it has long been suspected by astute clinicians, is both methodological and due to conceptual presuppositions. Thus, the review aspects of this article necessarily have an epistemological and historical bent. Also, it will be necessary to provide evidence of highly complex neural oscillators intrinsic to the basal ganglia–thalamic–cortical systems. Further, these oscillations are tied to the orchestration of motor unit behaviors that necessarily mediate the motoric effects of DBS. Next, the power of DBS to induce oscillations is discussed. Finally, preliminary evidence of complex interactions between the DBS oscillator and intrinsic oscillators in the subthalamic nucleus is presented.

The authors ask the indulgence of the reader. This is not a systematic review of all the publications that address the issue of DBS frequency. Such an effort is far beyond the opportunity reasonably offered by the editors. Thus, it is impossible to acknowledge the individual contributions made by many clinicians and scientists. This should not be taken as a judgment of their contributions or as a slight of any kind, rather a practical necessity. Second, this is a critique in the robust sense of the term. The critique recognizes that the issue of DBS frequency-based mechanisms is far from a complete understanding. Rather, considerable more research and scholarly efforts will be necessary. The strong critique is not a matter of fault-finding but more an effort so that future scientists and clinicians do not make the same mistakes that these authors and others have made. The term “error” as applied to any finding, inference, or conclusion is not a pejorative term. It is a wise person who learns from their mistakes; it a wiser person who learns from the mistakes of others. If one wants to change the future then one must see the present and the best way to see the present is to clearly see the past.

No theory or hypothesis arises spontaneously but rather is the consequence of a long history of reasoning. Reasoning laboriously worked through, perhaps centuries ago—for example by Aristotle (384–322 B.C.E.), now may be forgotten or taken for granted as a presupposition but is no less relevant. This fact was not lost on some of the greatest neuroscientists, such as Sir Charles Sherrington. Any competent critique must trace the conceptual antecedents as far back as is necessary to fully inform the review. It would be naïve, at best, or petulant, at worse, to discount analyses and wisdom from any age.

## 2. Conceptual History

Historically, the study of the effects of DBS frequencies has dichotomized the frequencies into high and low. This article demonstrates that such dichotomization is the result of a methodological error reinforced by highly intuitive conceptual appeal, also probably in error. This is not to say that clinicians did not see a difference when they stimulated at high frequencies compared to low frequencies. Rather, one error lies in undersampling in the frequency domain, resulting in aliasing. This is not a matter of opinion but rather a direct consequence of the Nyquist theorem. Another error lies in the pooling of results across subjects when the intersubject variability is high. The result is information loss according to the Second Law of Thermodynamics. The study by Huang et al. [3] demonstrates the effects of these errors.

Results of the study by Huang et al. [3] demonstrate that a number of DBS frequencies over a wide range for any individual subject resulted in improved bradykinesia as measured by hand opening–closing (Figure 1). If only a few DBS frequencies are studied, the result would likely demonstrate aliasing, and it will appear as though the relationship between improvement and DBS frequency is a monotonically increasing function, which it is not in reality. Further, there was considerable intersubject variability, suggesting that pooling subjects would have obscured the true relationship between improvement and DBS frequency.

The implications of the study by Huang et al. [3] go beyond the methodological (epistemological) to the reality (ontology) of the physiology of relevant parts of the nervous system, such as the basal ganglia–thalamic–cortical system. It is at least interesting that many years have passed since Cooper and colleagues’ description of DBS as presently implemented in 1980 [4] and popularized by Benabid and colleagues in 1987 [5] before the issue of DBS frequency was evaluated systematically over a range of frequencies. One wonders why.

Perhaps the need to study the range of frequencies systematically did not raise sufficient concern. However, this could only be the case if the prior concept of a dichotomous effect of DBS frequency was thought to be sufficient. Further, there is nothing in past observations that would be evidence that such a dichotomous effect is sufficient. At the very least, it could demonstrate the Fallacy of Limited Alternatives, where one explanation is considered more certain (falsely) when some alternatives are eliminated or not considered. However, for the one explanation to be certain, all reasonable alternatives would have to be considered and excluded, which, in the case of the effects of DBS frequencies, was not done.

Perhaps it was that the dichotomization of the effects of DBS frequencies seemed valid because of the analogy between pallidotomy and DBS in the vicinity of the globus pallidus interna. The logical fallacy of this analogy has been discussed elsewhere [6]. Interestingly, the contemporaneous theory of the pathophysiology of movement disorders, at theory positing excessive or reduced activity in the neurons of the globus pallidus interna (Globus Pallidus Interna (GPi Rate Theory)), was used to argue in favor of the analogy [7]. Considerable evidence shows that this notion of pathophysiology is no longer tenable [8].

The concerns raised here are more than historical. Successor theories likely share the same fatal flaw as the GPi Rate theory and therefore are at risk of delaying better theories. Current theories that posit excessive oscillations in the beta frequency range or excessive synchronization, along with the GPi Rate theory, share one-dimensional push–pull dynamics. At the very least, alternative theories that do not share these dynamics should be considered, particularly in view of the risks associated with the Fallacy of Limited Alternatives.

Given the lack of direct evidence, the presupposition of a dichotomous effect of DBS frequency then appears to be a default. Actually, dichotomization of phenomena and inferred causes marks human reasoning for millennia. Dichotomization is a consequence of the human epistemic condition. The inherent tendency to such a dichotomy, in appropriate circumstances, represents a pitfall only avoided if appreciated. Below the case is made for the ubiquity of dichotomization and, ultimately, its relevance to the consideration of DBS frequencies.

These dynamics were codified by Aristotle in his notion of *contraries*, where Aristotle wrote:
“The physicists (…) have two modes of explanation. The first set make the underlying body one—either one of the three or something else which is denser than fire and rarer than air—then generate everything else from this, and obtain in multiplicity by condensation and rarefaction. Now these are contraries which may be generalized into ‘*excess and defect*’ (italics added)”.[9]

Aristotle’s notion was extended by Galen (129–c. 200/c. 216, C.E.) in his notion of one-dimensional push–pull dynamics of relative excess or deficiencies of the humors. Galen’s dynamics continue underlying neurological and psychiatric therapeutics [10]. John Hughlings Jackson (1835–1911), one of history’s preeminent and defining neurologists, held a notion of positive and negative symptoms mediated by facilitation, inhibition, and inhibition of inhibition to release facilitation, continues to underlie mainstream neurological thinking. As neurology was the source of neuroscience, historically, it is not unexpected that the conceptual foundations of neurology would influence those of neuroscience. This inheritance is evident in the facilitation and inhibition of a hierarchy of reflex mechanisms in the work of Sir Charles Sherrington (1857–1952) in his Integrative Action of the Nervous System [11]. The one-dimensional push–pull dynamics are apparent in the descriptions of Phineas Gauge, where damage to the frontal lobes impaired the ability to suppress antisocial behavior, although the source or mechanism that produces antisocial behavior, thought to be released, has never been made clear.

Needed now is a Kuhnian paradigm shift [12] away from paradigms that presuppose one-dimensional push–pull dynamics. Despite celebrations written in *Science* and *Nature* 50 years after publication of *The*
*Structure of Scientific Revolutions* [12], Kuhn’s work continues to invite critique, fairly, but outright rejection by reasonable thinkers would be to make an oxymoron of the latter. Kuhn’s work was first and foremost a historical analysis. The uneven progress of science, even to the point of “getting stuck,” is a matter of historical fact. Such is the case as it relates to current theories of the pathophysiology of the basal ganglia–thalamic–cortical system as it informs hypotheses of the mechanisms of action of DBS, particularly as it relates to frequency.

Thomas Kuhn (1922–1996) argued that a paradigm shift occurs when observations unexplainable by the dominant paradigm—anomalies—accumulate to some breaking point. Kuhn left unexplained what the dynamics of the breaking point were, but many critics of science argue that the breaking point is a polemical issue. At what point do editors of scientific journals and grant administrators stop being accepting of failed paradigms? For example, almost since the inception of the GPi Rate theory, there have been contrary observations and anomalies, including the production of Parkinsonism in nonhuman primates with *N*-methyl-4-phenyl-1,2,3,6-tetrahydropyridine, that fail to demonstrate predictions of the GPi Rate theory [13], as well as demonstrating that pallidotomy improves hyperkinetic disorders contrary to the predictions. However, the GPi Rate theory persists even today, attesting to the ability of an intuitively appealing theory to trump fact [14].

## 3. Some Principles for Developing Alternative Theories

This critique is written with an eye to the future in the hope that much better hypotheses and theories will arise that, when vindicated, will extend our knowledge. There are some generic principles that may be helpful. Any theory is more than a set of facts and includes hypotheses that bridge gaps (interpolate) and extrapolate from observations to predictions. Hypotheses invoke notions of necessity and causality that go beyond correlations. However, the generation and evaluation of hypotheses in themselves are poorly understood and have been explained as happenstance, psychological, or aesthetic. However, Aristotle and other ancient Greek philosophers offered an important requirement called the Principle of Causational Synonymy that imposes constraints on any hypothesis and theory that posits some causal mechanism, such as changes in neuronal activities that cause, denigrate, or restore normal behavior. The Principle of Causational Synonymy holds that the means by which a cause acts to generate an effect must match the means in the effect that generates an effect. For example, when one pushes an object, the electrons in the outer orbit of the atoms on the surface of the hand repel the electrons in the other orbit of the atoms on the surface of the object. In the case of understanding the motor effects of DBS, these effects are through the recruitment and de-recruitment of motor units, as is discussed latter. As per the Principle of Causational Synonymy, no theory of the mechanisms of action of DBS can be considered complete or satisfactory without full explication of motor unit activities. Indeed, the inability of any theory or explanation to do so is evidence of serious shortcomings.

The Principle of Causational Synonymy can be extended to a Principle of Informational Synonymy. Considering information as nonrandom state changes, the nonrandom state changes, such as the frequency by which electrostatic charges are placed, reversed, and then stopped on the electrical contact of the DBS lead (the cause), must be synonymous with the neuronal changes in the vicinity of the DBS target that generate the DBS effect. Thus, both with the Principle of Causational Synonymy and the Principle of Informational Synonymy, there must be precise mapping of the dynamics of the causal agent with the effector agent.

Consider the situation of a person pushing a child on a swing. In addition to the repulsive forces in the appropriate electrons in the person and the child, there also is informational content in terms of the movements of the person pushing and the child swing. This is evident in the fact that the person pushing cannot push at any random time. Rather, the person pushing must be in phase with the child’s swing. Further, the pushing and swinging must be at the same frequencies, consistent with the Principle of Information Synonymy.

Applying these principles to the theories of DBS mechanisms of action, particularly as it relates to the stimulation pattern, in this case frequency, shows that current concepts are inadequate. Start with the effect, in this case the normalization of movement in the case of DBS for Parkinson’s disease. Ultimately, any changes in the movement must be implemented by orchestrating the recruitment and de-recruitment of motor units over a number of muscles. This orchestration is very complex and operates over multiple levels over varying time scales. This includes the recruitment of motor units by size. Reciprocal coordination exists between muscles agonistic and antagonistic to the intended joint rotation. Synergistic coordination of motor unit recruitments and de-recruitments in muscles spanning the same joint can be extended to motor unit orchestration simultaneously over multiple joints. Each of these operates at different time scales, and if related to oscillators in the nervous system (as will be demonstrated), then each time scale implies multiple oscillators at corresponding frequencies. Further, the operations at different frequencies are organized simultaneously. These issues are addressed in detail in Montgomery [15].

The fact that DBS improves motor control at specific frequencies argues that the frequency of the DBS is interacting in a nonrandom oscillatory manner with analogous oscillators within the basal ganglia-thalamic-cortical system. Indeed, as will be argued for, DBS can be considered as introducing an additional oscillator into the network of oscillators that comprises the basal ganglia-thalamic-cortical system. As discussed later, these oscillators are a specific kind that endows them with very important and interesting properties. The fact that multiple DBS frequencies exist shows that there are multiple oscillators of corresponding frequencies within the basal ganglia-thalamic-cortical system.

A possible alternative theory is presented here. The purpose here is to highlight alternative conceptions and not necessarily to champion any one particular theory. This is consistent with the intent of providing a critique rather than a review. To the authors’ knowledge, the proposed theory is rather novel, at least in literature searches of various databases, thus providing a study of contrast. For example, there is no other publication searchable in PubMed that discusses motor unit recruitment abnormalities in Parkinson’s disease or the effects of DBS. Nor are there any publications on discrete neural oscillators. Understandably, research bearing on the alternative theory largely will be those pursued by the authors. The theory advanced here includes the following:
The basal ganglia–thalamic–cortical system is involved in the recruitment and de-recruitment of motor units through oscillators within the system that then drive the motor cortex and brainstem structures that project to the lower motor neurons of the spinal cord and brainstem.The basal ganglia-thalamic-cortical system is composed of multiple loosely coupled re-entrant, nonlinear polysynaptic discrete oscillators (Figure 2). Note that the term “discrete” is emphasized, as nearly all discussions of oscillators within the context of DBS, physiology, and pathophysiology do not make a distinction between continuous harmonic oscillators and discrete oscillators. These two types of oscillators have very different properties and dynamics. This theory was described previously in general terms [16] and in more detail in Montgomery [1].Oscillatory states consist of oscillators representing different frequencies simultaneously and that oscillatory states can shift dynamically, in the manner of a bifurcation in Complex Systems.DBS acts as a noisy discrete oscillator when introduced into the basal ganglia-thalamic-cortical system, which interacts with oscillators inherent to the basal ganglia-thalamic-cortical system.Interaction of the DBS oscillator within the oscillatory network is a model of mechanisms inherent in the basal ganglia-thalamic-cortical system, both in normal and in pathological circumstances. These interactions include the following:
Positive and negative stochastic resonanceNoisy synchronizationPhase control such as advancement and entrainment

## 4. Evidence for a Role of the Basal Ganglia-Thalamic-Cortical System in Motor Unit Recruitment and De-Recruitment

Oscillator activities in motor unit behaviors have been known, and perhaps underappreciated, since described by Piper in 1907 [17]. At high muscle contractions, a 40-Hz signal can be appreciated, not only by electromyographic (EMG) recordings but also by auscultation using a stethoscope [18]. The genesis of such oscillatory activity is not known but likely is of central nervous system origin [19,20]. The question arises whether the basal ganglia via the motor cortex plays a role in these motor unit oscillations specifically, as suggested by abnormalities of the Piper rhythm in patients with Parkinson’s and motor unit orchestration generally.

Motor unit recruitment normally is organized based on the muscular force at which a motor unit becomes active. The phenomenology of motor unit recruitment order is the recruitment of small motor units with low force requirements followed by a recruitment of larger motor units with a greater force requirement and is known as the Henneman Size Principle. Small motor units generated are thought to provide the small forces necessary for fine resolution of the forces generated, whereas large motor units supply the strength.

Another important extrapolation from the phenomenology is to the underlying mechanisms. Initially, it was hypothesized that motor unit recruitment was determined by the local biophysical properties of the lower motor neuron with large lower motor neurons, corresponding to larger motor units, being harder to excite. Subsequently, differential synaptic inputs with respect to lower motor neuron size, particularly peripheral afferents, were postulated to play a role. Nevertheless, most studies have not discussed or have discounted the role of descending inputs. Indeed, Enoka writes, “The great advantage of a spinally based control scheme (recruitment order determined by motor neuron biophysics—authors), such as orderly recruitment, is that it relieves higher centers (for example, the basal ganglia—authors) of the responsibility to select the motor neurons that must be activated for a specific task” [21]. Thus, it is not surprising that Parkinson’s disease’s effects on motor unit recruitment order have not received much attention.

In a study of motor unit recruitment [22], subjects were recruited from patients implanted previously with DBS systems in the vicinity of the subthalamic nucleus (STN). Following an overnight fast from their usual medications used to treat Parkinson’s disease, subjects placed their extended finger and wrists into a manipulandum through which they could exert an isometric flexor force about the wrist. The task was to exert a force starting at rest to their maximum voluntary force over 60 s. Fine wire hook electrodes were inserted into the flexor carpi ulnaris. An example is shown in Figure 3. A raw EMG is shown in the top panel of Figure 3 under the condition of therapeutic DBS in the vicinity of the subthalamic nucleus. The therapeutic DBS electrode configuration and stimulation parameters were those determined optimal during prior routine clinical care with frequencies at least greater than 100 pps. As can be appreciated readily, there is a step-like increase in the amplitudes of the motor unit potentials with increasing force, clearly consistent with the Henneman Size Principle. However, under the condition of no DBS in the same subject, the step-like increase is not seen. Further, large amplitude motor unit potentials are seen early in the task at low forces, clearly inconsistent with the Henneman Size Principle.

The EMG signals associated with individual motor units were extracted using a template-matching algorithm [23] to identify individual waveforms. Subsequently, another computer program used these waveforms as the basis to decompose superimposed signals into the elemental waveforms, allowing discrimination of individual motor units to higher percentages of maximal force production. Curve fitting of a sigmoidal function to cumulative discharge histograms over the ramp force allowed determination of the force at which the motor unit was recruited. The force at recruitment was correlated with the motor unit size determined by the area under the curve of the motor unit potential waveform (Figure 4). A positive slope indicates consistency with the Henneman Size Principle.

As can be seen in Figure 4 for a subject with Parkinson’s disease, the slopes are near or less than 0 with the stimulator off or at 20 pps. With higher frequency DBS, the slopes become more positive. Indeed, the therapeutic DBS frequency resulted in the greatest positive slope. The reasonable conclusion is that the Henneman Size Principle is violated without treatment but is restored with higher frequencies of DBS. Figure 5 shows results comparing slopes for patients in the DBS Off (0 pps) and DBS On (therapeutic pps) for another patient. As can be seen, there was a near reversal of the normal recruitment pattern. Large motor units were recruited at the lowest forces generated and the small motor units only at the highest force. The motor recruitment changed to what would normally be expected with therapeutic DBS. Slopes under the different conditions for 14 subjects with Parkinson’s disease and 10 normal subjects are shown in Figure 6.

The question is what accounts for restoration of the normal motor unit recruitment profile with high-frequency therapeutic DBS and the dependence on frequency as shown in Figure 4. Changes in the recruitment order may be mediated by the motor cortex, whose upper motor neurons project to the lower motor neurons of the motor unit. Studies in nonhuman primates and in humans demonstrate a short latency highly temporally consistent with antidromic activations in motor cortex neurons in response to DBS pulses applied in the vicinity of the subthalamic nucleus. Evidence also shows an antidromic activation of the cortex in response to DBS in the vicinity of the subthalamic nucleus in humans with Parkinson’s disease [24]. However, there were subsequent responses at longer latencies that suggest the possibility of re-entrant oscillations. Such an effect is demonstrated more clearly in the recording of neurons in the ventral thalamus pars oralis in response to DBS in the vicinity of the globus pallidus interna (Figure 7). The buildup of responses at approximately 5 ms following the DBS pulse suggests a resonance effect.

One possible explanation is that an action potential generated in the axon of the thalamic neuron passing in the vicinity of the DBS contact proceeds in an antidromic manner, as demonstrated in the recording of extracellular action potentials (Figure 7), as well as orthodromically to the cortex. There, the stimulated motor cortex neurons sent action potentials back to the thalamic neurons. The timing is such that the action potentials generated in the thalamic neurons consequent to driving by cortical neurons collided with the antidromic action potentials. This process could be modeled as the activity traversing one-half of the pathway from the thalamus to the cortex and the entire pathway from the cortex to the thalamus. Estimating from the latency of the resonant buildup of 5 ms, the total transit time in the thalamic–cortical feedback loop would be on the order of 6.7 ms. This would correspond to a frequency within the thalamic–cortical feedback loop of approximately 150 Hz.

These considerations suggest that one possible mechanism for motor improvements with DBS in the vicinity of both the subthalamic nucleus and the globus pallidus interna may reflect resonance interactions between the oscillator composed of the thalamic-cortical loop and the oscillator in the manner of DBS. The resonant interactions between the DBS oscillator and the thalamic-cortical oscillator are thought to improve the signal-to-noise ratio by stochastic resonance, thereby reducing the misinformation exiting from the motor cortex to the lower motor neurons and restoring the normal order of motor unit recruitment.

## 5. Stochastic Resonance

The next question is whether the model described here could account for the improvement in bradykinesia at other DBS frequencies, as demonstrated in Figure 1. The basal ganglia-thalamic-cortical system can be considered a network of interconnected polysynaptic re-entrant nonlinear discrete oscillators (Figure 2). There are a number of different oscillators with differing numbers of nodes; thus, the DBS at several different frequencies could interact via stochastic resonance with any of the multiple oscillators. Evidence in support of this hypothesis is given here.

Improvement in motor control can be considered increased information in the effectors, which is in the orchestration of motor unit recruitments and de-recruitments. Recordings of extracellular action potentials were obtained as a nonhuman primate preformed an arm-reaching task. Peri-event rasters and histograms show neuronal activities before and after a behavioral event (Figure 8). As can be seen, under the condition of no stimulation, there is no change or modulation of the neuronal activities. However, with DBS-like stimulation in the vicinity of the subthalamic nucleus at 150 pps, there is a consistent modulation of the neuronal activities with the behavior. The modulation was less at 100 pps and still less at 50 pps. Again, it is not likely that the information represented by the modulation of neuronal activities at 150 pps DBS originated in the DBS pulse train (a violation of the Principle of Informational Synonymy). Rather, modulation was present there, although not observable over the “noise”. Thus, the signal, being the underlying modulation of neuronal activities, was not seen above the noise; in other words, there was a poor signal-to-noise ratio. The reverse has been observed, in which a signal in the neuronal responses with no DBS was progressively lost with higher DBS frequencies (Figure 9). The DBS at 150 pps affects the signal relative to the noise, but in different directions and dependent on DBS frequency.

If this model holds for DBS, then the observation that multiple but specific DBS frequencies improve hand opening-closing bradykinesia argues for multiple oscillators involved in the hand opening-closing behavior. Indeed, it is likely that there is an orchestration of motor unit recruitment and that de-recruitment operates over multiple time scales, each related to an oscillator within the basal ganglia-thalamic-cortical system [15].

## 6. Evidence for Oscillations in the Basal Ganglia–Thalamic–Cortical System

The Schuster periodogram [26] is an alternative to a Fourier transform of detecting periodic (oscillatory) activity in a time series. A variant of the Schuster periodogram has been demonstrated as mathematically equivalent to the Fourier transform but is easier to implement (Figure 10) [27]. This method was used to analyze neuronal spike trains from neurons in the basal ganglia-thalamic-cortical system from two nonhuman primates at rest [28]; an example of the spectrogram is shown in Figure 11. The color of the pixels in the image represents the *z*-score difference from the same spike train where the interspike intervals (ISI) were randomized. A 2-s window of the spike train was sampled and the *z*-score change was calculated. The window was then moved over the entire spike train at 0.2-s steps.

As can be seen, there are multiple and high-frequency oscillatory activities in the neuronal spike train simultaneously. This multiplicity of frequencies and changes in the sets of frequencies can be seen in the following: 24 neurons were recorded in the globus pallidus externa, 15 in the globus pallidus interna, 16 in the putamen, 49 in the sensory cortex, 9 in the subthalamic nucleus, and 25 in the motor cortex [28]. The average frequency for each structure ranged from 135 to 140 Hz. Mathematical modeling demonstrates that the individual spike trains in loosely coupled nonlinear oscillators operating at noncommensurate frequencies can entrain multiple frequencies simultaneously [1,29].

The theory is offered that each oscillator of different frequencies affects different physiological mechanisms over different time scales, for example, the orchestration of motor unit recruitment and de-recruitment as described previously. Thus, it would not be unexpected that DBS at different frequencies would have multiple effects on motor behavior as shown previously in Figure 1. However, it is not just any frequencies but specific and precise frequencies, perhaps reflecting the specific frequencies entrained in the neuronal spike trains of the basal ganglia-thalamic-cortical system.

## 7. Evidence for DBS-Induced Oscillations

### 7.1. Direct Observations

Studying the effects of DBS in the vicinity of the subthalamic nucleus in a nonhuman primate on neuronal action potentials provided an example where DBS could induce oscillations in the basal ganglia-thalamic-cortical system [28]. An example is shown in the peri-event raster and histogram of the time interval between DBS pulses for a cortical neuron (Figure 12). As can be seen, there are three peaks in the post-DBS response with 50 pps DBS. This would correspond to a frequency of oscillation of 150 Hz. Two peaks are noted in the interval between pulses with stimulation at 100 and 130 pps. Of note, the second peak at 100 pps occurs sooner that at 50 pps and the second peak is even earlier at 130 pps. These findings could be explained if the DBS pulses induced an inherent oscillation of approximately 150 Hz. Thus, the inherent frequency would resonate with DBS at 50 pps. However, the effect at 100 and 130 pps would be a combination of the response driven by the DBS pulses at those frequencies and the inherent oscillations phase entrained at 50 pps DBS. Nonetheless, this observation supports the hypothesis that DBS can induce ongoing oscillations.

### 7.2. Resonance Effects to Pair-Pulse Stimulation

The basic concept is that if the basal ganglia-thalamic-cortical system represents a closed loop or feedback system of oscillators, then a single pulse might reverberate in an oscillatory manner through the loop. If a second pulse is timed precisely to the returning effect from the prior pulse, these effects should be additive (Figure 13). Thus, an experiment was conducted using pairs of DBS pulses. The first pulse was hypothesized to initiate a re-entrant oscillation, and the second pulse was timed to interact with the re-entrant activity generated by the first pulse. The time interval between the first and the second pulse of the pair corresponds to the cycle time of the oscillator.

Two nonhuman primates (*Macaca mulatta*) were trained using only positive reinforcement to sit quietly in a loosely restraining chair and to allow passive movement of their arms. After training, a recording chamber was implanted surgically over a craniotomy site such that microelectrodes could be passed through the intact dura into the basal ganglia as described previously [30]. The protocol received prior approval by the Institutional Animal Care and Use Committee of the Cleveland Clinic Foundation.

A reduced scale model of the human DBS lead (NuMed Inc., 2880 Main St. Hopkinton, NY, USA) was placed such that the deepest contact was at the bottom of the subthalamic nucleus using stereotactic methods confirmed by microelectrode recordings of extracellular action potentials. Locations of the DBS-like leads, as well as other microelectrode recordings, were later confirmed by histological analyses. The leads had four contacts, each 0.525 mm in diameter, 0.5 mm long, and with a 0.5-mm space between contacts, giving a total surface area of 0.82 mm^2^ per contact. Using biphasic square-wave pulses, constant current electrical stimulations, using biphasic square-wave pulses, were delivered using the most distal contact (contact No. 0) referenced to the most proximal contact (contact No. 3). The morphology of the pulses was such that, at contact No. 0, the cathodic phase preceded the anodic phase. The reverse was true at contact No. 3. A pulse width of 90 μs per phase was used. The amplitude selected was 80% of the current that produced tonic contraction, presumably from the current spread to the internal capsule.

Standard methods of microelectrode recordings of neuronal extracellular action potentials were used. As is standard in electrophysiological studies, autocorrelograms for all neurons were inspected for the absence of a refractory period as evidenced by a high probability of a neuronal discharge within 3 ms of the index discharge. Lack of a refractory period would indicate that the waveforms isolated and attributed to a single neuron were an admixture of two or more neurons. Data from these neurons were discarded. Also, all pair-wise cross-correlograms for neurons isolated from a single microelectrode recording site were constructed. Any cross-correlograms showing a refractory period effect were interpreted as the two neurons actually represented a single neuron that was isolated incorrectly. If found, data for these pairs of neurons were pooled. Loss of signal related to the effects of the stimulator artifact was accounted for and corrected.

Continuous recordings of neuronal activity before and during trains of paired-pulse stimuli were made. Resonance effects were demonstrated by constructing post-stimulus rasters, and histograms that were indexed to the second of the pulse pair, test pulse. Rasters and histograms were constructed by taking segments of data immediately after the second or test pulse for 20 ms (Figure 14). This corresponds to the period between each pair of pulses. Post-stimulus histograms were constructed where each sequential time bin (0.4 ms) contained counts of neuronal discharge for the 20 ms that followed the second test pulse. These time bins were then averaged across the number of stimulus pairs applied, giving a neuronal discharge probability in 0.4-ms time increments following the second test pulse.

For comparison across different neurons and different structures, these post-stimulus histograms of neuronal discharge probabilities were normalized by expressing the neuronal discharge counts in bins as a *z* score based on the mean and standard deviation of neuronal discharges during the baseline prior to stimulation. Because of the short interstimulus intervals, it was not possible to compare neuronal activities following the first pulse to activities following the second pulse. Sampling of the prestimulation period values was determined by creating virtual stimulation pulses so that neuronal activity during the prestimulation period could be analyzed exactly as the neuronal activity during the stimulation period (Figure 14). This was chosen as a means to sample the prestimulation baseline randomly.

Although the interstimulation pulse intervals were not random and therefore the virtual stimulation train was not random, neuronal activity in the prestimulation period is independent of the subsequent virtual stimulation pulse train and therefore can be considered random with respect to the time of neuronal discharge. The mean and standard deviation of the neuronal discharge probability in the time bins during the prestimulation baseline were calculated as described previously for the stimulation period. In this case, histograms were indexed to the second virtual pulse. The probability of neuronal discharge in each time bin in the post-actual stimulus histogram was converted to a *z* score by subtracting the mean discharge probability per bin in the post-virtual stimulation histogram and dividing the difference by the standard deviation of the bin probabilities in the post-virtual stimulation histogram. This method was chosen over other techniques such as absolute or percent change in discharge frequency because (1) the *z* score normalizes data, thus allowing comparisons between different neurons in different structures; and (2) the *z* score accounts for the variability of neuronal discharge activity, thus allowing inferences as to statistical significance.

One hundred and eighteen neurons were recorded in the basal ganglia-thalamic-cortical system in two nonhuman primates (in one nonhuman primate, only 15 neurons studied were in the motor cortex). Forty neurons were recorded in the motor cortex, 25 in the somatosensory cortex, 16 in the caudate and putamen, 14 in the globus pallidus internal segment, and 23 in the globus pallidus external segment. While the number of neurons analyzed within each structure was relatively small, the intent was to study the dynamics of the basal ganglia-thalamic-cortical system such that consistency of observations across different nuclei was the primary interest. As such, the number of neurons for the system studied is similar to other published studies. Analyses within each structure were secondary.

An example of a response in a motor cortex neuron is shown in Figure 15. Each colored bar represents a change in the probability of a neuronal discharge after the second or test pulse compared to the prestimulation baseline associated with a *z* score equal to or greater than 1.96. As can be seen, the significant resonance effects for longer interstimulus intervals had different latencies than the response to the interstimulus interval of 1 ms. Consequently, it is not likely that these other responses were mediated by the same mechanism or that they represent a direct response to the second pulse. Figure 15 also shows that this neuron had multiple resonance effects at longer latencies that were associated with 4-, 5-, 7-, and 8-ms interstimulus intervals, corresponding to a resonance frequency of 250, 200, 143, and 125 Hz, respectively. Interestingly, such high-frequency oscillations also have been noted in local field potential recordings in the subthalamic nucleus of patients with Parkinson’s disease through implanted DBS leads [31]. However, inferences from local field potentials are problematic given that these represent the summed activities of a large number of dendrites, some of which may be irrelevant yet interact to affect the local field potential in any event.

Results shown in Figure 15 suggest evidence of a periodic excitability arising within the single neuron. The first response with the 1-ms interstimulus interval pulses occurs 3.8 ms after the test pulse, suggesting that this was not a direct antidromic effect, but probably that the neuron being analyzed was at least one synapse from the neuron being stimulated. Interestingly, after the 1-ms interstimulus interval test pulse, the neuronal discharge probability increased significantly at 7.6 and 15.2 ms during the 20 ms after the test pulse; these probably are harmonics of the initial latency of 3.8 ms. Analyses were conducted on 118 neurons, all of which showed at least one resonance effect with paired-pulse stimulation at different interstimulus intervals. The median number of significant resonance frequencies (25th to 75th percentile; number of neurons) in the pooled caudate and putamen was five (2 to 7.5, *n* = 16); globus pallidus externa, 9 (5.5 to 11.75, *n* = 23); globus pallidus interna, 6 (1 to 9, *n* = 14); somatosensory cortex, 7 (3.5 to 8.5, *n* = 25); and motor cortex, 7 (4.5 to 12, *n* = 40). Thus, most neurons had multiple resonance effects at different frequencies.

## 8. Evidence of DBS Oscillatory Interactions with Intrinsic Oscillators within the Basal Ganglia-Thalamic-Cortical System

Recent studies demonstrate antidromic action potentials in the subthalamic nucleus neuronal in response to DBS in the vicinity of the contralateral STN [32]. Interestingly, only a small percentage of DBS pulses resulted in antidromic action potentials. This result is consistent with other studies [32,33,34,35,36]. One wonders whether this result indicates a purely stochastic process or some underlying determining mechanism such as the specific dynamics in the neuronal membrane potentials in the soma. STN neuronal microelectrode recordings were used in subjects with Parkinson’s disease made during DBS in the vicinity of the contralateral STN in a manner described elsewhere [32]. Stimulation was at 160 and 30 pps. Fifty-eight neurons were recorded from eight STNs in eight subjects.

Spike trains containing only antidromic action potentials (antidromic-only spike trains) were constructed from the original spike trains by retaining only those time stamps of antidromic action potentials. Fifty-two of the 58 neurons demonstrated antidromic action potentials. Randomized antidromic-only spike trains were created by shuffling the order of the antidromic action potentials in the antidromic-only spike trains. This was accomplished by dividing the antidromic-only spike trains into segments between successive stimulation pulses. Each interstimulus pulse interval may or may not have an antidromic action potential.

The hypothesis to be addressed is that sequence of antidromic action potentials are not random. Thus, it is necessary to compare the actual sequence of antidromic action potentials to what would be a random sequence. The orders of the consecutive interstimulus pulse intervals were randomized (shuffled) in order to affect a breaking up of any periodicity in intervals containing an antidromic action potential, creating a randomized antidromic-only spike train. If the probability of an antidromic action potential is not random, the interspike interval histogram of the randomized antidromic-only spike train would differ from the ISI of the corresponding antidromic-only spike train.

Periodicity in the antidromic-only spike train was examined by comparing the antidromic-only spike train to the randomized antidromic-only spike train using power spectral densities (PSDs). It is important to note that the antidromic action potentials are time-locked to the stimulation pulse. The frequencies of the antidromic action potentials thus stand in complex relation to the DBS frequencies.

Figure 16 and Figure 17 show representative antidromic-only spike trains and randomized antidromic-only spike train ISI histograms for two neurons for DBS at 160 and 30 pps, respectively. As can be seen, an antidromic action potential occurs only after a stimulation pulse. The time resolution is thus the interstimulus pulse interval. Importantly, the varying magnitude of the peaks suggests that different probabilities of an antidromic action potential follow different stimulation pulses. Such would not be the case, however, if every stimulation pulse was associated with an antidromic action potential or if the probability of an antidromic action potential was random relative to the stimulation pulses. Rather, there appears to be “structure” in the probabilities of an antidromic action potential. This is further supported by the lack of peaks at such times when the sequential order of the antidromic action potentials is randomized. Of the 52 neurons demonstrating antidromic action potentials studied, 24 showed a significant difference between the antidromic-only spike train and the randomized antidromic spike train ISI. These latter neurons received additional study.

Power spectral densities (PSDs) were constructed on antidromic spike trains from antidromic-only spike train and randomized antidromic-only spike train data. Figure 18 shows a representative example. A strong peak appears at 66 Hz, a lesser peak at 26 Hz, and a small peak appears at 92 Hz in the antidromic-only spike trains. This latter peak is not present in the randomized antidromic-only spike train. The clear distinction in these peaks from the rest of the frequencies in the PSD attests to their significance. Figure 19 shows the PSD for the same neuron that appears in Figure 6 but does not appear under the 30-pps DBS condition. Peaks are observed at 30 Hz (DBS frequency), 26 Hz, 22 Hz, 6 Hz, and 3 Hz. Original spike trains demonstrated no peaks in the PSD of neuronal activity during the baseline before DBS; a representative example is shown in Figure 20.

The various peaks are listed in Table 1. Oscillations at 66 and 26 Hz occurred independently of other frequencies, which suggests that these oscillators are fundamental to the subthalamic nucleus. In addition, the 26-Hz oscillations were seen at both 160 and 30 pps DBS. These are not a consequence of the stimulation frequency. They likely represent, rather, a fundamental oscillator involving the subthalamic nucleus. Also, the peaks at 26 and 66 Hz are not harmonics of the DBS frequencies. Interestingly, several neurons had both 66- and 26-Hz oscillators. Other peaks probably represent beat frequencies that result from the interaction of the fundamental frequencies and the DBS frequency. For example, peaks detected at 92 Hz probably are the result of interactions between the 66-Hz fundamental oscillator and the 160-pps DBS (the actual stimulation frequency was measured at 157.5 pps). As can be seen, the 92-Hz peak was never present without the peak at 66 Hz. Similarly, the 3-Hz peak likely represents the beat frequency from the interaction of the fundamental 26-Hz frequency and the 30-pps DBS, as the 3-Hz peak was never seen at the 160-pps DBS and was never independent of the 26-Hz peak under the 30-pps DBS. By virtue of the fact that the 30-pps DBS is closer to the fundamental frequency at 26 Hz than the 160-pps DBS, which would produce high-frequency harmonics and beat interactions, other peaks under the 30-pps DBS likely represent beat interactions among harmonics of the DBS and the fundamental frequencies.

The relatively few neurons available for analyses mean that any conclusions are tentative, and the difficulty of these experiments makes it unlikely that significantly more data are forthcoming. As one awaits future confirmation or refutation, these remarkable findings—their implications in particular—warrant reporting, albeit with the appropriate caveats kept in mind.

The study led to the following conclusions: For many neurons, occurrence of an antidromic action potential is not random. The probability of an antidromic action potential reflects the underlying state of the neuronal excitability, which means that the antidromic action potential can be used as a probe of the underlying neuronal dynamics. The probability of an antidromic action potential and the underlying neuronal state of excitability appear periodic or oscillatory. Not simple harmonics of the DBS frequency, the frequencies of the underlying oscillatory neuronal states are, rather, independent of the latter. One cannot exclude the possibility that the actual frequency of these oscillators is not some integer multiple of the 26- or 66-Hz oscillators. The beat frequencies of 3 and 92 Hz for some of the neurons suggest that 26 and 66 Hz are the fundamental frequencies for the involved oscillators. The key finding for the purpose of this article is that DBS can be considered a discrete nonlinear oscillator, which interacts with independent oscillators within the subthalamic nucleus, possibly arising from oscillators intrinsic to the basal ganglia-thalamic-cortical system.

## 9. Computational Simulations of the Basal Ganglia-Thalamic-Cortical System as a Network of Loosely Coupled, Nonlinear Re-Entrant Discrete Oscillators

As a proof of concept, although not of biological fact, computational simulations were conducted using an architecture derived from the theory that the basal ganglia-thalamic-cortical system is organized as large, loosely coupled, nonlinear polysynaptic discrete re-entrant oscillators (Figure 21), details of which are discussed elsewhere [1,16]. A detailed description of discrete neural oscillators can be found in Montgomery [1]. Briefly, a node is defined as a collection of local neurons that generally share the same inputs and project to same neurons in another node. For example, a node may be confined to a single anatomical structure and each anatomical nucleus (and region of cortex) consists of a number of nodes. An oscillator consists of a set of nodes whose neurons are connected with previous and subsequent nodes of neurons. During any cycle of reentrant activities within the oscillator, a subset of neurons within each node may discharge sufficient to maintain the oscillations yet not leading to saturation or collapse of the oscillator. The simulations are designed to determine whether neural oscillators of the general architecture shown in Figure 21 are capable of sustained oscillatory activities, what would be the nature of those activities and what if any similarity of the oscillations in the simulation to analyses of neuronal activities recorded in human and non-human primates.

The operations of neurons in the simulation are depicted in Figure 22. Each neuron is of the integrate-and-fire type. Postsynaptic potentials resulting from synaptic inputs are modeled as decaying exponential functions. The output is a discrete pulse representing an action potential based on the summed synaptic inputs exceeding a threshold. The threshold is dynamic so as to account for changes in ionic conductances with subthreshold changes in the neuronal membrane potential, for example, depolarization blockade (elevated threshold) and post-hyperpolarization rebound (decreased threshold). A probability function determines whether an action potential arriving at the presynaptic terminal induces a postsynaptic potential change and reflects the known inefficiency of synaptic transmission. Conduction times and synaptic delays were established based on studies described previously.

One instantiation of the computational modeling is shown in Figure 23. There are three interconnected oscillators with different numbers of nodes and thus different inherent fundamental frequencies. The inherent frequency is defined as number of times a “bit of information”, such as an action potential, can traverse the oscillator in one second and depends on the number of nodes within the oscillator. A spectrogram showing the frequency contents over time for a representative neuron in the computational network is shown in Figure 24, along with spectrograms obtained from a globus pallidus externa of a nonhuman primate and from a neuron recorded in the subthalamic nucleus of a human with Parkinson’s disease. As can be seen, the outputs are very similar and consist of multiple frequencies entrained simultaneously, sets of frequencies that are stable over brief periods of time between bifurcations.

## 10. Conclusions

The physiological mechanism underlying the effect of DBS frequency remains an enigma, particularly as the complexities of the relationship between the DBS frequencies and motor effects are more fully recognized. However, addressing the enigma is a great opportunity because, typically, new knowledge is needed in order to solve it. New knowledge can only come from new hypotheses, which most often require new perspectives. New perspectives require letting go of old perspectives, at least those aspects that are counterproductive. This requires exercise in epistemological analysis as well as experiments.

It is likely that the effects of DBS frequency have a great deal to say about the underlying dynamics of the pathophysiology and physiology of the basal ganglia-thalamic-cortical system. A better understanding of those dynamics could provide a metaphor for research that extends beyond the basal ganglia-thalamic-cortical system.

## Figures and Tables

**Figure 1 brainsci-06-00034-f001:**
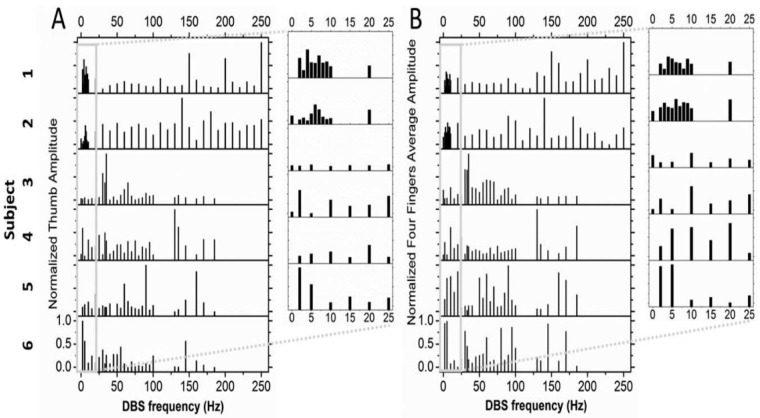
Patients with Parkinson’s disease executed rapid hand opening and closing while wearing a glove that recorded finger and thumb movements. Following an overnight fast, these patients received their first dose of medications used to treat Parkinson’s. Movements of the fingers were separated from those of the thumb. The movement amplitudes were normalized by scaling from 0 to 1 where 1 represents the maximum amplitude. The mean over all the movements was analyzed: (**A**) for the thumb and (**B**) for the fingers (pooled across all the fingers) for each subject. Absence of a column indicates that the associated stimulation frequency was unavailable with the subject’s Implanted Pulse Generator (IPG). Multiple peaks in amplitude movements are found across multiple frequencies, including low frequencies. (Inserts) Amplitudes for the lower range of stimulation frequencies are shown in the expanded window [3].

**Figure 2 brainsci-06-00034-f002:**
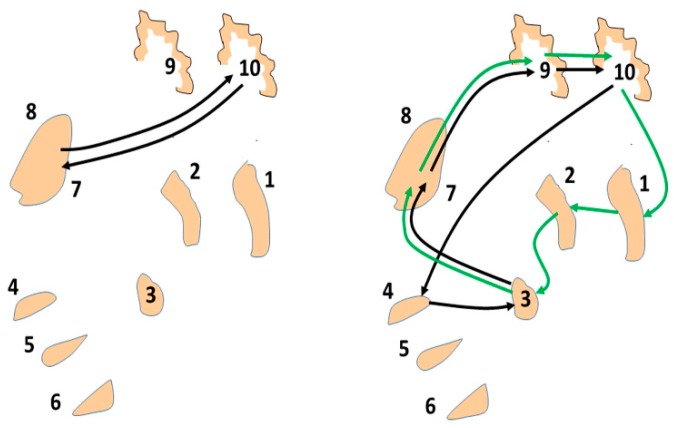
Schematic representation of the various components of the basal ganglia–thalamic–cortical system, demonstrating three oscillator architectures out of a greater number of possible oscillators. Represented are the following various structures: 1, putamen (as representative of the striatum); 2, globus pallidus externa; 3, globus pallidus interna; 4, subthalamic nucleus; 5, substantia nigra pars reticulata; 6, substantia nigra pars compacta (location of the cell bodies that utilize dopamine as their neurotransmitter); 7, ventral thalamus pars oralis; 8, parafasicular and centromedian nuclei of the thalamus; 9, supplementary motor area; and 10, primary motor cortex. Note that the GPi participates in two different oscillators, each of which is associated with a different fundamental frequency—a five-node loop and a six-node loop [1].

**Figure 3 brainsci-06-00034-f003:**
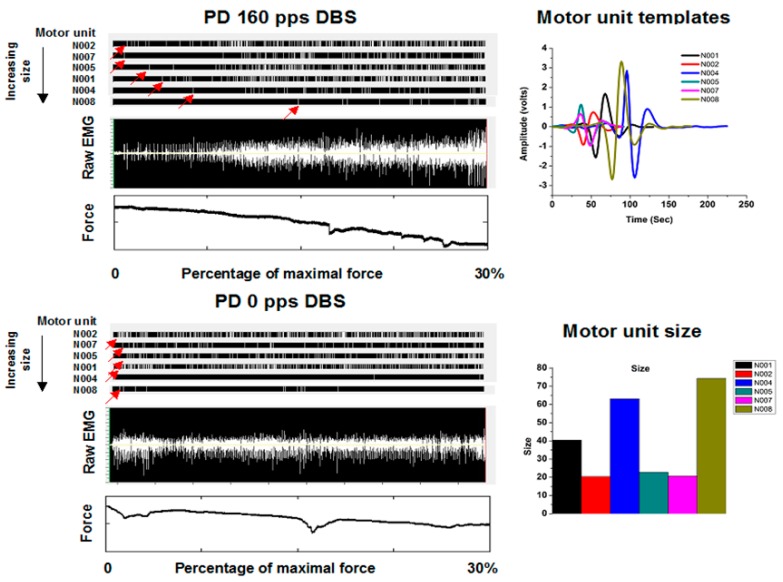
Representative example of raw intramuscular electromyographic (EMG) activities in a subject with Parkinson’s disease under conditions of 160 pulses per second (pps) DBS (therapeutic) and 0 pps DBS. The raw EMG under 160 pps DBS has a stair-step appearance. Each step is associated with recruitment of a larger motor unit. The same six motor units were identified under both conditions and their waveforms and size are shown. The waveform associated with each motor unit is distinct and varies in size from its fellows. The size was determined by measuring the area under the curve. The time of occurrence of a motor unit discharge appears in the raster, with one row for each motor unit. The red arrow indicates onset of activities as the force generated increases progressively. In the 160-pps DBS condition, an orderly recruitment is shown (indicated by red arrows): Consistent with the Henneman Size Principle, smaller units are recruited first, followed by progressively larger motor units. Under the 0-pps DBS condition, the orderly recruitment of motor units is lost and the units are recruited nearly simultaneously. Large motor units are recruited early in the task and at small forces.

**Figure 4 brainsci-06-00034-f004:**
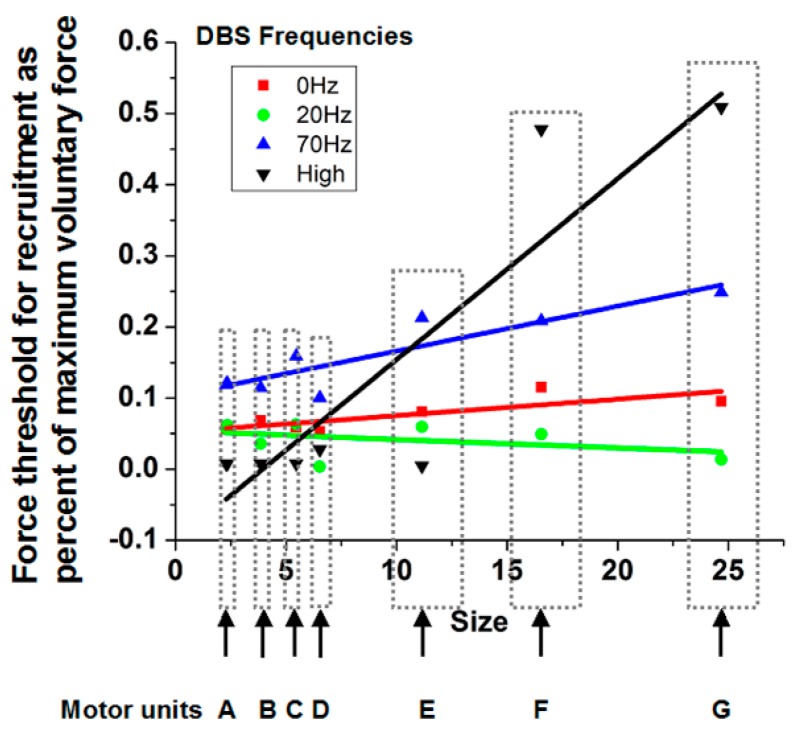
Relationship between percent of maximum force at which each of the seven motor units were recruited and the size of the motor units for different DBS conditions for a representative subject with Parkinson’s disease. The seven motor units (A–G) are ordered according to motor unit size: A is the smallest and G is the largest. Different symbols and colors represent the various DBS conditions. A flat or small slope indicates that the Henneman Size Principle did not hold under that DBS condition. An initially relatively flat slope for the untreated patient (0 Hz DBS) increases greatly with therapeutic (high) DBS, suggesting that motor unit recruitment has become normalized.

**Figure 5 brainsci-06-00034-f005:**
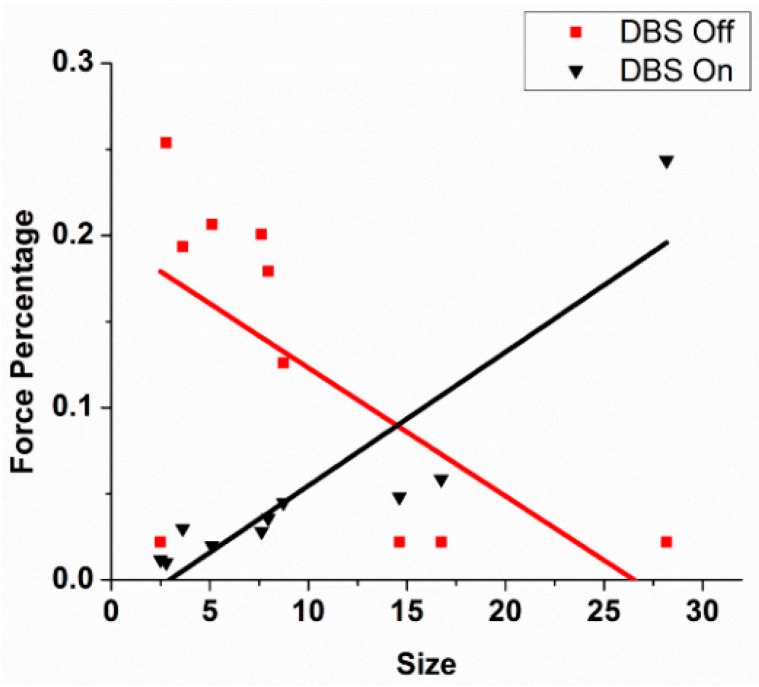
An example of the relationship between the size of the motor units and the percentages of maximum voluntary force at which the motor unit was recruited. In the DBS Off condition, the recruitment is opposite that predicted by the Henneman Size Principle. The recruitment became what would normally be expected by the principle.

**Figure 6 brainsci-06-00034-f006:**
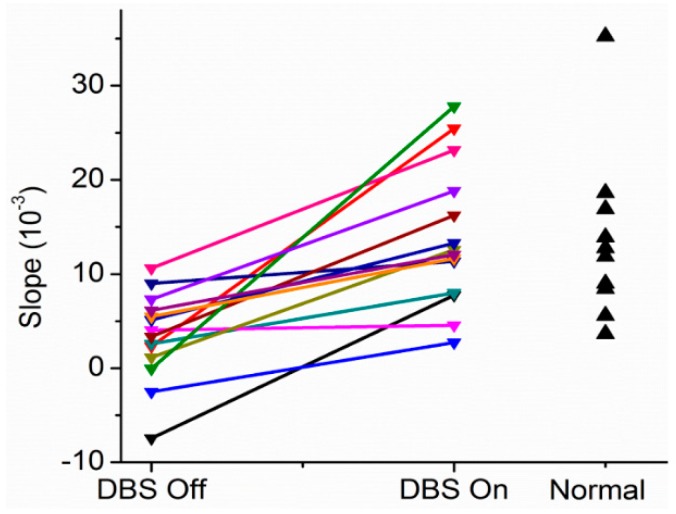
Summary of the changes in slopes of the motor unit recruitment order for 14 patients off and on therapeutic DBS compared to 10 normal controls.

**Figure 7 brainsci-06-00034-f007:**
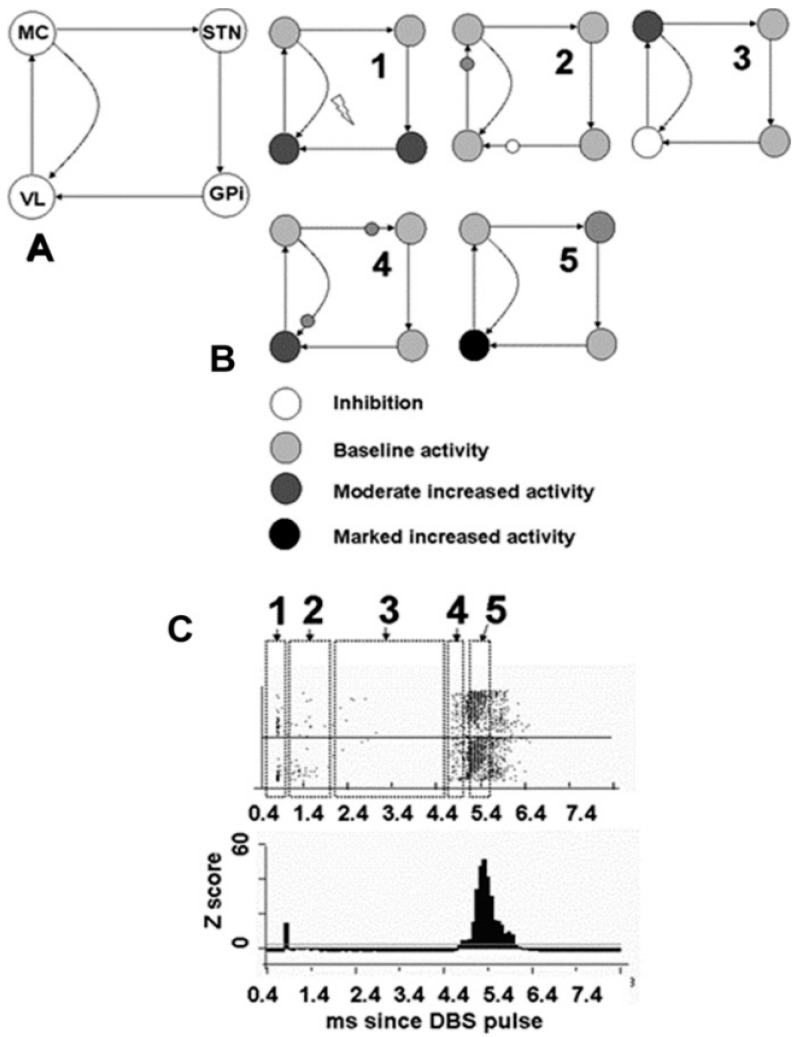
Some ventral thalamus pars oralis neurons (designated VL) demonstrate a remarkable posthyperpolarization (postinhibitory) rebound increased excitability (**C**). A potential mechanism is represented schematically (**B**); a nested two oscillator system is shown (**A**). The first oscillator is the disynaptic feedback loop between the motor cortex (MC) and the ventral thalamus pars oralis, the basal ganglia relay nucleus of the thalamus. The second loop consists of the MC to the subthalamic nucleus (STN), globus pallidus interna (GPi), Vop, and motor cortex once again. Each numbered step (**B**) shows subsequent activations, which begin with synchronized activation of the ventral thalamus pars oralis and GPi neurons in step 1. The activity in Vop is then transmitted to MC, and the activity in GPi is transmitted to the ventral thalamus pars oralis in step 2. This results in excitation of the MC and inhibition of the ventral thalamus pars oralis (VL) in step 3. MC activity is then transmitted back to the ventral thalamus pars oralis coincides with a posthyperpolarization rebound-increased activity in Vop following GPi axonal influences on the thalamic neurons in step 4. Excitation from the MC in step 4 combines with the postinhibitory rebound-increased excitability in the ventral thalamus pars oralis to produce a marked increase in activity, as shown in step 5 (modified from Montgomery [25]).

**Figure 8 brainsci-06-00034-f008:**
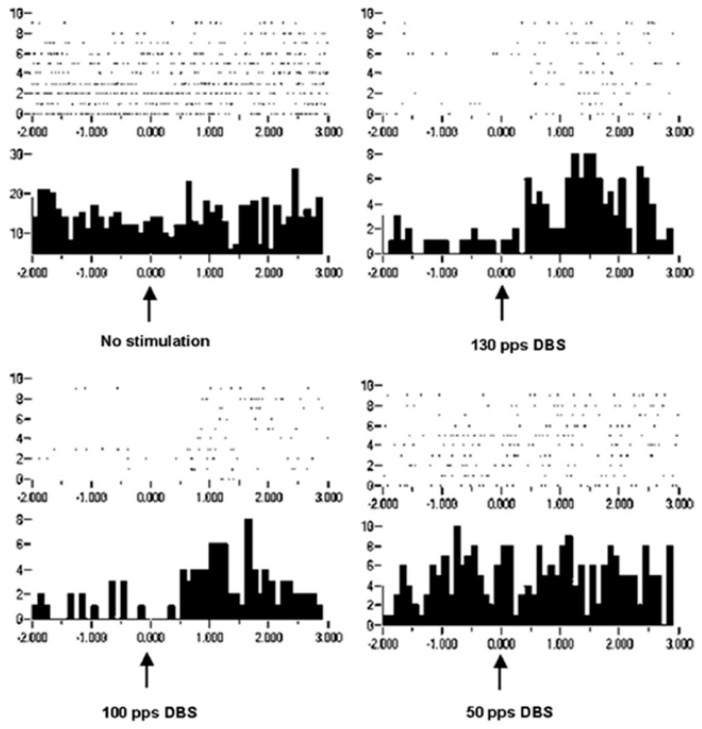
Peri-event rasters and histograms for a neuron recorded in the putamen of a non-human primate. Each dot in the rasters represents a neuronal discharge and each row represents a trial. Summing a column in the raster produces the histogram. There is no meaningful modulation of neuronal activity with behavior (appearance of the commencement signal at time zero is indicated by the up arrow) under the no stimulation condition. However, consistent modulation occurs at 130 pps and, to a lesser extent, at 100 pps DBS, suggesting that the DBS has enlisted the neuron into being meaningfully related to the behavior. It bears noting that the baseline activity prior to the commencement signal is reduced [25].

**Figure 9 brainsci-06-00034-f009:**
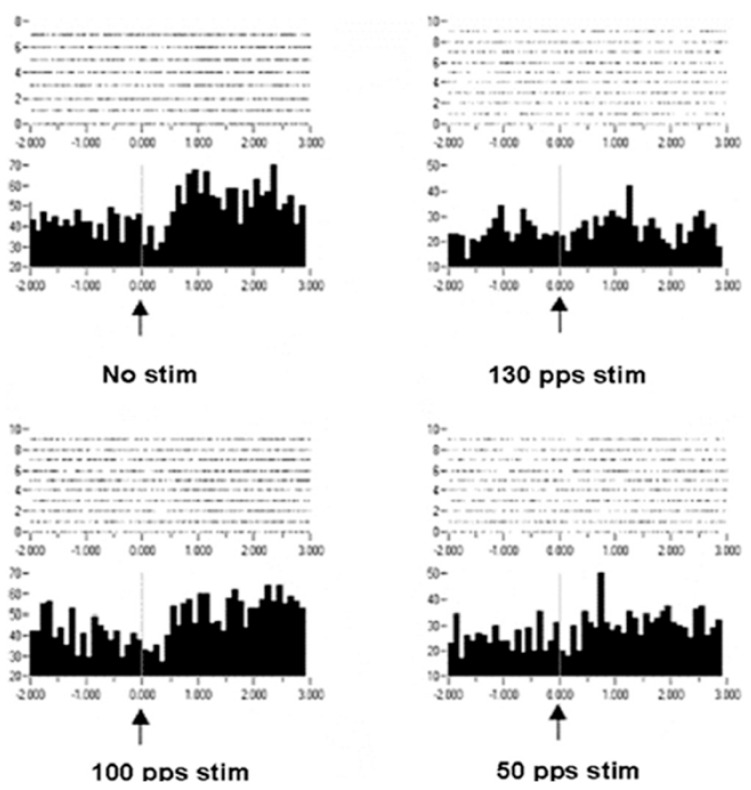
Peri-event rasters and histogram showing activity of a caudate nucleus neuron before and after onset of the commencement signal, indicated by the up arrow. Each dot in the rasters represents a neuronal discharge and each row represents a trial. Summing a column in the raster produces the histogram. With no stimulation, with 100 pps DBS, and, to a lesser degree, with 50 pps DBS, there is an increase in neuronal discharge following the onset of the commencement signal. This dynamic modulation is lost with the 130-pps DBS [25].

**Figure 10 brainsci-06-00034-f010:**
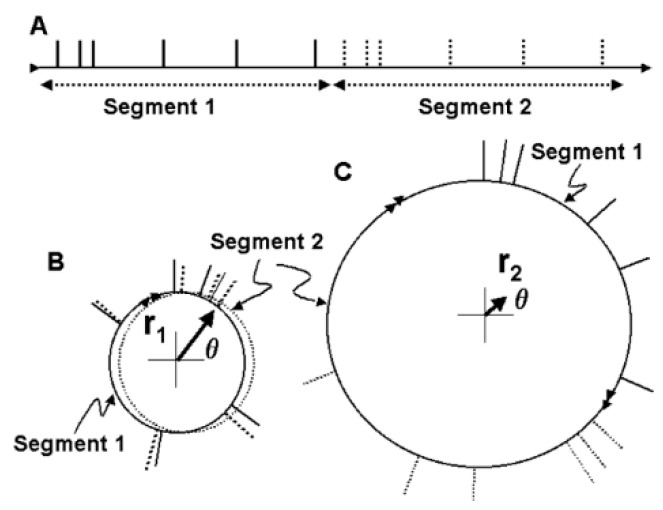
Schematic representation of the Schuster periodogram utilizing circular statistics. (**A**) A spike train is divided into segments; (**B**) each of the two segments is closed to form a circle, and the time of occurrence of each action potential is plotted as a unit vector on the circumference. As can be seen, the unit vectors are not distributed randomly over the circumference. A resultant vector is determined (r_1_), which, in the situation depicted by B, will be nonzero. The magnitude of the resultant vector indicates the statistical significance of the power at the frequency corresponding to the lengths of the segments. The resultant vectors can be calculated for segments of any size, thus corresponding to a specific frequency analogous to the power at any frequency in a Fourier transform. The angle, *θ* indicates the phase of the periodic or oscillatory activity.

**Figure 11 brainsci-06-00034-f011:**
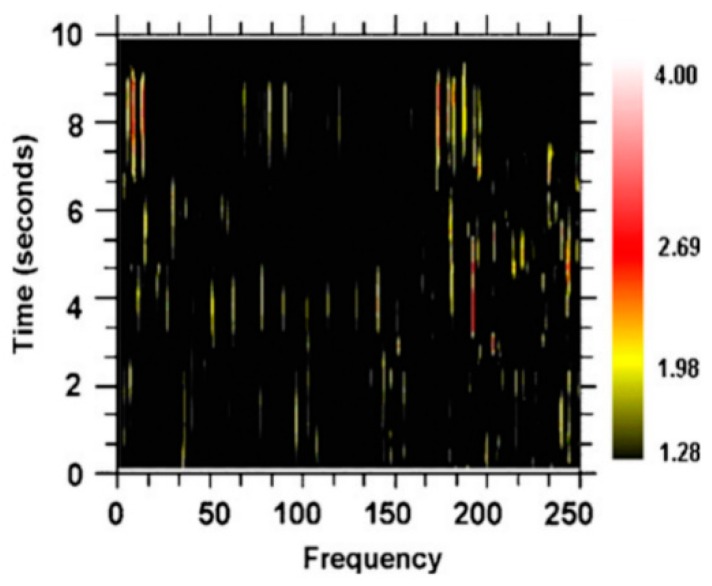
Spectrogram relative power indicated as a *z*-score change over the randomized spike train using a 2-s window moved through time in 0.2-s steps. As can be appreciated, many frequencies are represented simultaneously in the neuronal spike train. Further, it appears that there are sets of frequencies that change over time, suggesting bifurcations from metastable states associated with each set of frequencies [25].

**Figure 12 brainsci-06-00034-f012:**
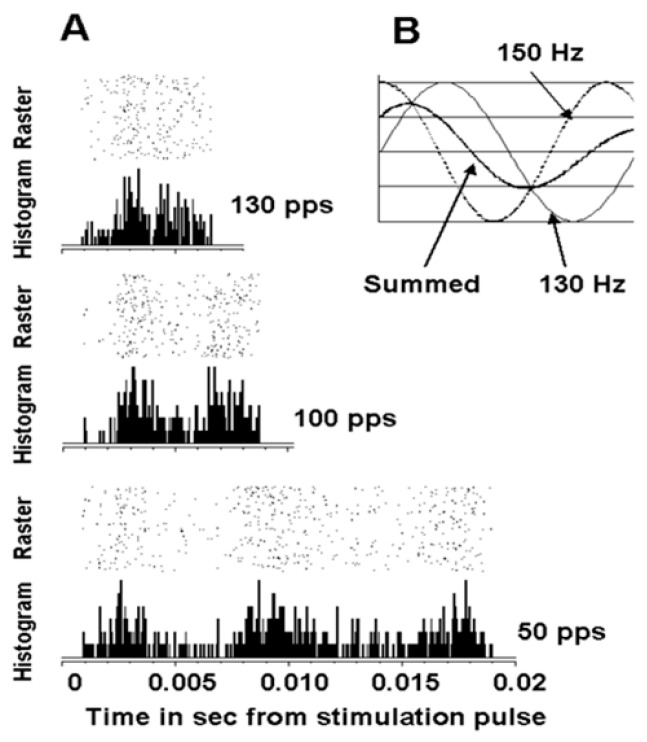
(**A**) A post-stimulus raster and histogram of a cortical neuron recorded in a nonhuman primate using a 50-pps DBS-like stimulation delivered in the vicinity of the STN. The stimulus pulse occurred at time 0, and the raster and histogram show the periodic neuronal activities during the interstimulus pulse intervals. Each dot in the raster represents the onset of an action potential. Each row represents responses to a single DBS pulse. The number of dots in columns across the raster appears in the histogram below each raster. The lengths of the raster and histogram represent the time interval between DBS pulses. In the raster and histogram associated with the 50-pps DBS, a recurring peak of increased neuronal activity is evident. The truncated appearance of the third peak suggests that the time period associated with the frequency of the re-entrant oscillatory activity is not an integer multiple of the time interval between the DBS pulses. This suggests an interaction between the oscillator generating the recurrent activity and the oscillator composed of the DBS spike train (**B**). This observation may be explained by an interaction between a 130-Hz DBS oscillator and a 150-Hz oscillator intrinsic to the neuron.

**Figure 13 brainsci-06-00034-f013:**
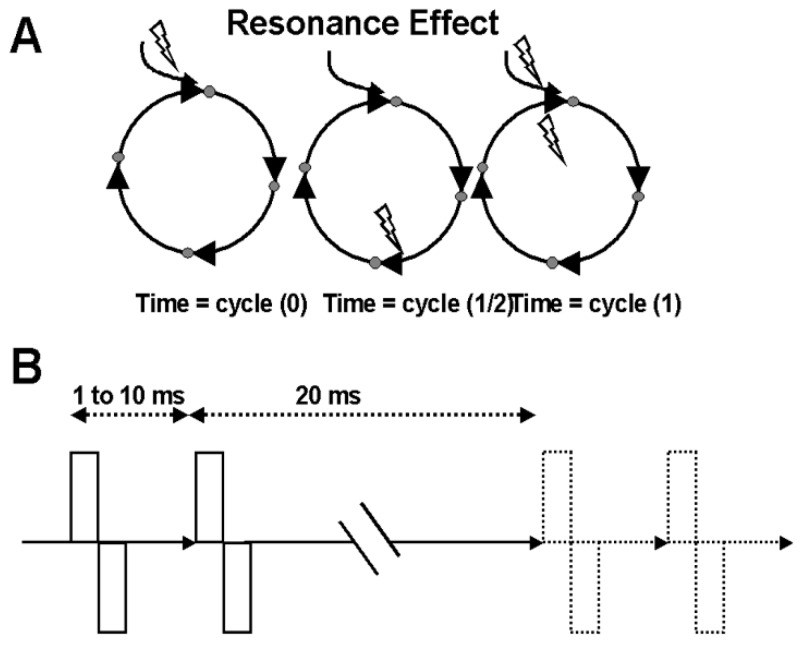
(**A**) A schematic representation of the resonance effect. The first stimulation (conditioning pulse) causes an excitation to traverse the closed loop. If the second stimulation (test pulse) is delivered just as the excitation effect from the first or conditioning pulse returns to the original site, the temporal summation on the neuronal cell membrane will amplify the response. (**B**) Paired-pulse stimulus trains represented schematically. The interstimulus interval represents a specific frequency (1/interval). This study examined the frequencies represented by the intervals from 1 to 10 ms (1000 Hz to 100 Hz) at 1-ms increments.

**Figure 14 brainsci-06-00034-f014:**
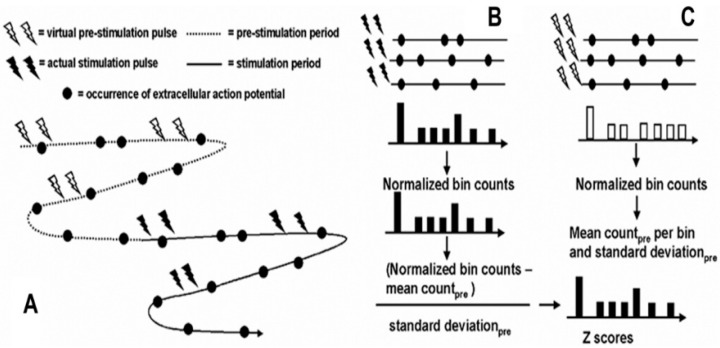
Schematic representation of analysis methods for detecting a resonance effect for paired-pulse stimulation. A set of virtual stimulus pulse pairs was created during the prestimulation period by translating the timing of the actual stimulation pulse pairs into the prestimulation period (**A**). Post-stimulus rasters and histograms were constructed indexed to the second pulse of the actual (**B**) and virtual (**C**) stimulation pulses. The rasters were collapsed across rows into the time bins (0.4 ms) of the histograms, resulting in counts of extracellular action potentials. This was normalized by dividing by the number of sets of paired pulse stimuli, resulting in probabilities of neuronal discharge in each time interval following the second stimulus pair. The mean probabilities per bin and the standard deviation were calculated for the virtual stimulation histograms (**C**). The mean was then subtracted from each time bin probability during the actual stimulation and divided by the standard deviation, resulting in a *z* score (**B**).

**Figure 15 brainsci-06-00034-f015:**
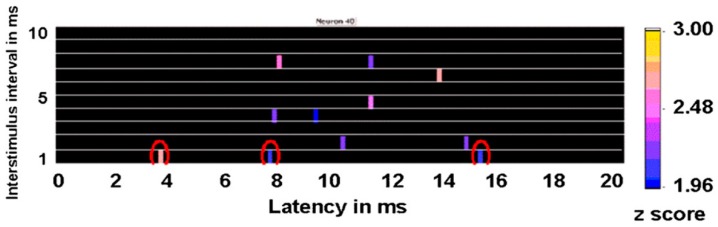
Results from paired-pulse experiments for a neuron recorded in the motor cortex of a nonhuman primate. Each row represents changes in the probability of a neuronal discharge from baseline for each interstimulus interval of the paired-pulse stimuli. Colored bars represent any change that has a *z* score greater than 1.96 compared to baseline. The horizontal axis represents the latency of the resonance effect after the second or test pulse of the pair.

**Figure 16 brainsci-06-00034-f016:**
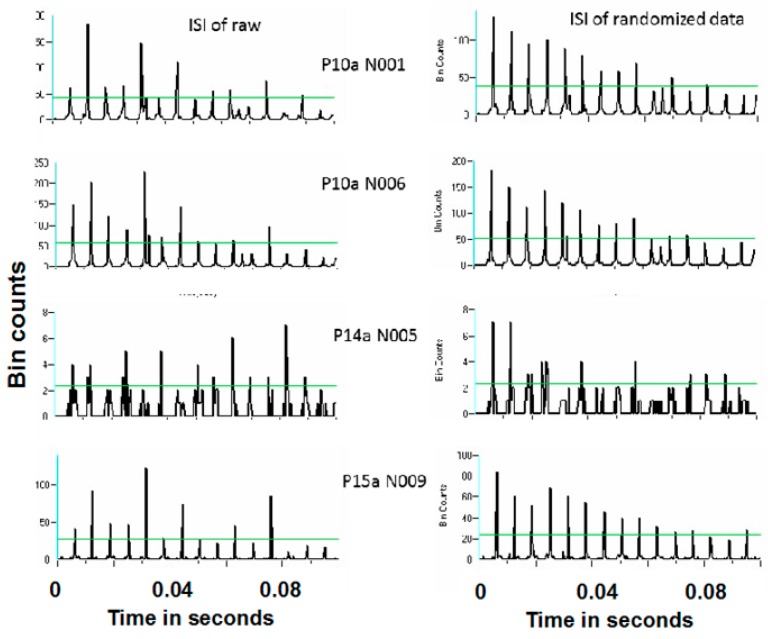
Representative interspike interval histogram of a antidromic-only spike train and its randomized-only spike train counterpart for 160 pps DBS.

**Figure 17 brainsci-06-00034-f017:**
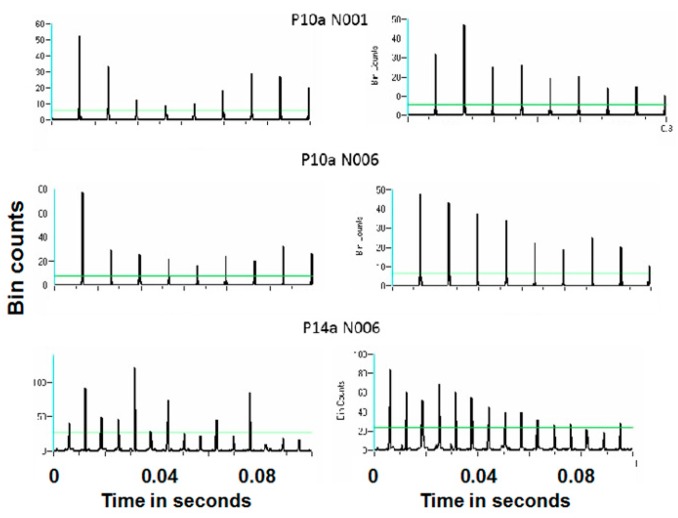
Representative interspike interval histogram of a antidromic-only spike train and its randomized-only spike train counterpart for 30 pps DBS.

**Figure 18 brainsci-06-00034-f018:**
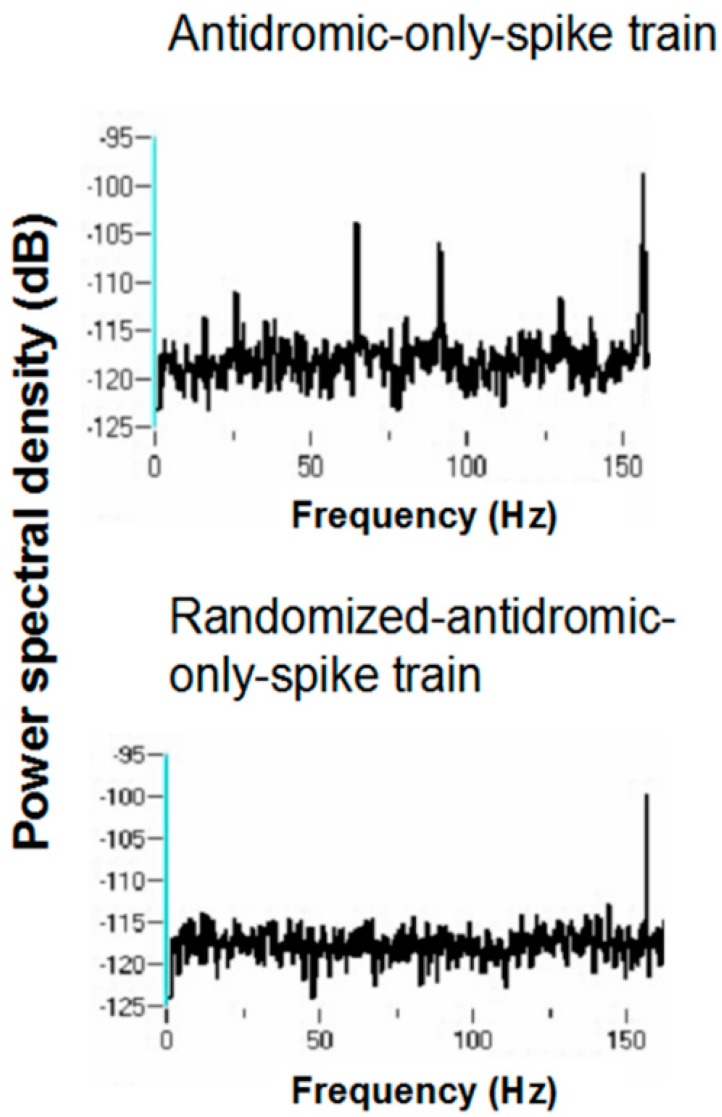
Representative power spectral density of a antidromic-only spike train and its randomized-only spike train counterpart for 160 pps DBS.

**Figure 19 brainsci-06-00034-f019:**
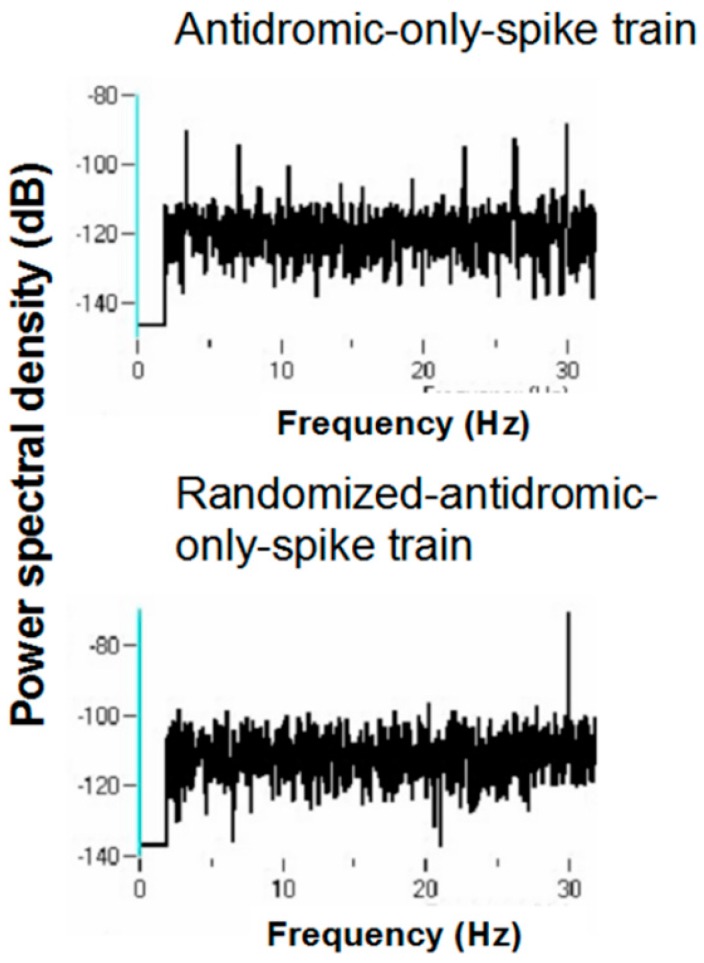
Representative power spectral density of a antidromic-only spike train and its randomized-only spike train counterpart for 30 pps DBS.

**Figure 20 brainsci-06-00034-f020:**
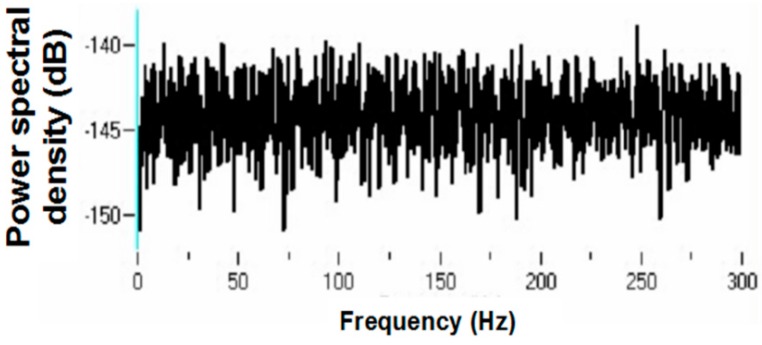
Representative power spectral density of an original train prior to DBS.

**Figure 21 brainsci-06-00034-f021:**
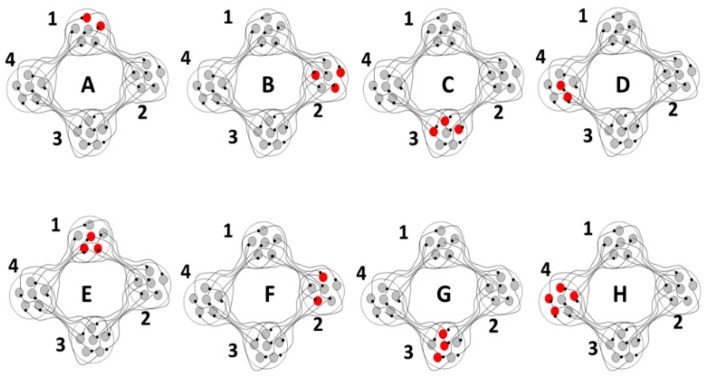
Schematic representation of the basic architecture of the computer simulations. Represented is one oscillator to show the structure of neurons within each node and how information is transmitted via subsets of neurons within each node. A four-node oscillator (nodes 1–4) whose nodes contain five neurons each is illustrated. A series of time intervals (A–H) are represented. Neurons become active in node 1 at time A. The activity then propagates to neurons in subsequent nodes. For any part of the cycle, a subset of neurons in each node becomes active and the specific neurons that are active vary with each cycle. Thus the neurons active at time B in node 2 are different than the neurons active in node 2 at time F. Multiple oscillators of different numbers of nodes, thus different inherent fundamental frequencies, can be linked [1].

**Figure 22 brainsci-06-00034-f022:**
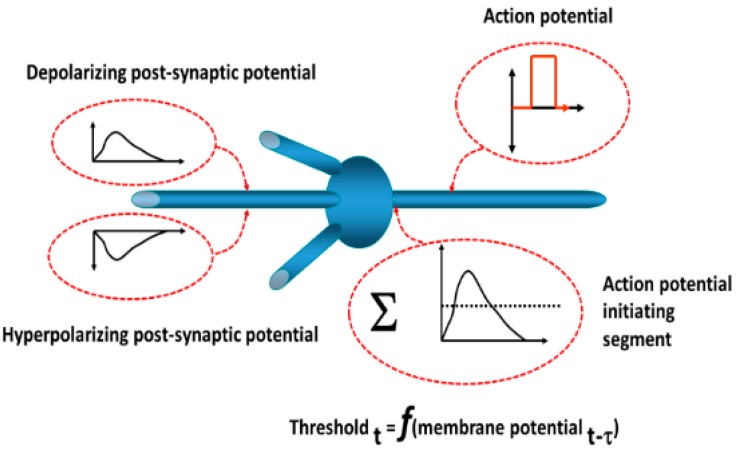
Model neurons consist of multiple dendritic inputs affecting the postsynaptic transmembrane electrical potentials. These inputs are summed at the action potential-initiating segment. If the summed potentials exceed a dynamic threshold, an action potential is generated. The postsynaptic potentials are modeled on as either a depolarizing or a hyperpolarizing decaying exponential function. The threshold, for example, at time *t_i_*, varies with the previous transmembrane electrical potential just prior at time *t_i-1_*. This allows changes in Na^+^ voltage-gated ionic conductance channel activation and inactivation. In this manner, post-hyperpolarization excitation and depolarization blockade are modeled [1].

**Figure 23 brainsci-06-00034-f023:**
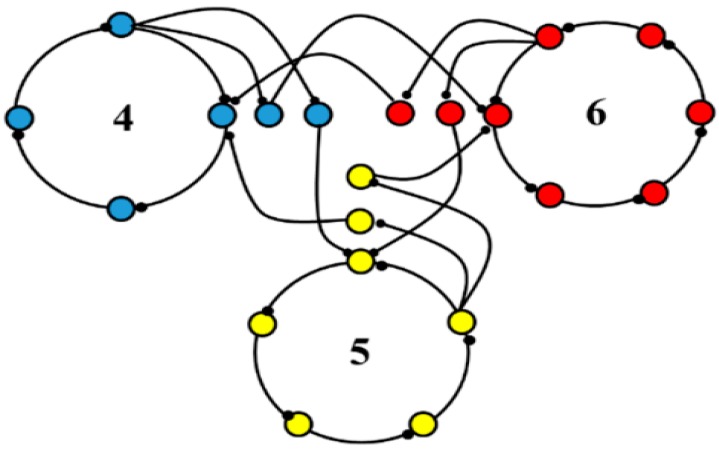
One instantiation of the four-oscillator network. There are three oscillators that consist of four, five, and six nodes, respectively. Each node contains 100 neurons. In each oscillator, a designative node is also connected to the corresponding node in the other oscillators, creating a loosely coupled network. The neurons in each node received inputs from all neurons in the previous node and sent outputs to each neuron in the subsequent node [1].

**Figure 24 brainsci-06-00034-f024:**
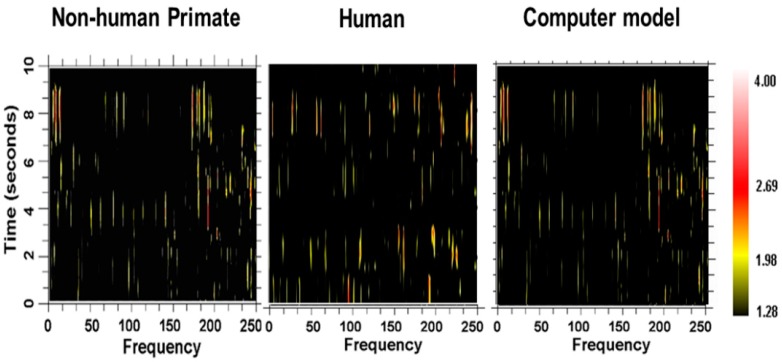
Spectrograms showing the amount of activity (power) at different frequencies over time. The power is represented as the *z*-score difference (color scale) over a randomized train of action potentials (spike train). At each point in time, the train of action potentials has a specific set of frequencies over a range of frequencies. Also, the train of action potentials appears to be stable in their frequency content and then change (bifurcate) to other sets of frequencies. Spectrograms of actual recordings in the globus pallidus externa of a nonhuman primate, the subthalamic nucleus in a human, and a neuron in the computer model of the network are shown.

**Table 1 brainsci-06-00034-t001:** Results of power spectral densities on antidromic-only-spike trains. Frequencies associated with the peaks are reported (in Hz).

Subject	Neuron	160 pps DBS	30 pps DBS
15a	N001	66	
	N002	66	
	N003	66	
	N004	66, 92	
	N006	66, 92	
	N007	66, 92	
	N008	66, 92	
	N009	26, 66, 92, 136	
	N014	26, 66, 92, 136	
p10a	N001	26, 66, 92, 136	3, 26, 23
	N002	26, 66, 92, 136	3, 26, 23,6
	N004	26	
	N005	26, 66, 92, 136	3, 26
	N006	26, 66, 92, 136	3, 26, 22, 6, 10, 19
	N011	26, 66, 92, 136	3, 26, 6, 22
	N019	26, 66, 92, 136	3, 26, 6, 22
p03a	N001	26, 66, 92	3, 26
	N005	26, 66, 92 136	3, 26
	N008	66, 92	
	N016	26, 66, 92 136	3, 26

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
