# Peer review of "Deep Brain Stimulation Frequency—A Divining Rod for New and Novel Concepts of Nervous System Function and Therapy"

_brainsci, 2016, doi:10.3390/brainsci6030034_

Round 1

Reviewer 1 Report

·      The tone of the article is fairly aggressive. There are claims that, essentially, the authors are the only ones to grasp the subtleties of DBS frequency and that most authors/investigators in the field do not think about the possible effects of stimulation other than classical high frequency. This is patently false. The works of Grill and Tass, among others, have considered these topics for years. Even those of us who have not published extensively on frequency are aware of some of these discordant observations. There may be a kernel of insight here, but it is hard to find among the repeated reminders that the rest of us are stupid and the authors are brilliant.

·      There is frequent appeal to philosophical concepts rendered in capital letters, to remind us that they are Great and Important Principles. This is unnecessary, and honestly detracts, because it adds to this overall tone problem. The article would be much clearer if the authors dispensed with Aristotle and claims that they (and only they) have the key to a "paradigm shift", and started more simply with statements of fact regarding evidence that supports vs. does not support the "high vs low" dichotomy. Basically, I would start this article with the material on its current Page 5.

·      If this is meant to be a critical review of the literature in this topic (see prior comment), primary data other than that of the authors' own papers should also be considered and presented.

·      The theory around line 332, that there is reason to suspect a specific interaction of an intrinsic motor circuit frequency and the preferred clinical DBS frequency, should really be developed much earlier in the article, and should be expounded upon further here.

·      The authors make a comment early on that a limited gridding of frequency leads to an under-appreciation of the effects of frequency. They then proceed to multiple examples where a handful of frequencies are used, generally spaced by tens of Hz. Figure 4 specifically labels a category of DBS as "high" without breaking it down further by granular frequencies, precisely what the authors claim the field has been doing wrong all these years. Where is the explanation for this apparent inconsistency?

·      There really needs to be some kind of table summarizing how the cited evidence base stacks up against various structures. The authors jump around to different basal ganglia structures in each example, making it hard to believe that they are finding general principles.

·      It is not clear how the primary data of Figure 15 were corrected for multiple statistical comparisons. (It is also not clear how those data support the idea of the reentrant loop.)

·      The captions/legends of Figures 16-17 are not at all clear to me. What is supposed to be different here? Why should randomizing the times of a spike train lead to a change in the ISI distribution? Are these subplots individual neurons, which is what the labels imply?

·      The article ends rather abruptly after a chunk of primary data, without ultimately summarizing what the authors' theory is and what they think should be done next.

·      There are typos, e.g. "widow" for "window", that should be ground out by further proofreading.

Broadly, at the end of reading this, I am left with a sense that I have been given a tour of the senior author's major papers, but with no clear unifying theory other than "things in the motor system oscillate, so DBS probably interacts with them". That is a perfectly valid theory, but it is not a paradigm shift, and it is not particularly falsifiable when given at this level of generality. This article needs to be re-written to clearly identify what has been done beyond the authors' own work, state their idea about a new model, then present the core components of that model.

Author Response

Comments and Suggestions for Authors

Reviewer 1 -   The tone of the article is fairly aggressive. There are claims that, essentially, the authors are the only ones to grasp the subtleties of DBS frequency and that most authors/investigators in the field do not think about the possible effects of stimulation other than classical high frequency. This is patently false. The works of Grill and Tass, among others, have considered these topics for years. Even those of us who have not published extensively on frequency are aware of some of these discordant observations. There may be a kernel of insight here, but it is hard to find among the repeated reminders that the rest of us are stupid and the authors are brilliant. 

MONTGOMERY – I am not sure how to respond to these criticisms.  What does it mean to say that the tone is “fairly aggressive”?  Nowhere in the article is there any suggestion that others, such as Grill and Tass, are stupid.  The comments suggest a very narrow perspective where disagreement with others is taken as an insult.  When is honest disagreement or differences in perspective mean that the authors are brilliant and those with different theories or perspectives are stupid?  We cannot cite every author who has mentioned “frequency” when writing on this subject but this appears to the reviewer as distain for those not cited.  I would hope that the reviewer would avoid the temptation to condemn some idea of an author just because he or she did not think of it first.  Reviewer 2’s criticisms are naïve and border on petulance.  Such an overtly hostile and antagonistic misreading of the paper clearly calls into question the objectivity of the reviewer and suggests such hidden agenda.  If two reviews are required, both should be reasonable.  We strongly encourage the editors to seek a more balanced review.

Reviewer 1 -   There is frequent appeal to philosophical concepts rendered in capital letters, to remind us that they are Great and Important Principles. This is unnecessary, and honestly detracts, because it adds to this overall tone problem. The article would be much clearer if the authors dispensed with Aristotle and claims that they (and only they) have the key to a "paradigm shift", and started more simply with statements of fact regarding evidence that supports vs. does not support the "high vs low" dichotomy. Basically, I would start this article with the material on its current Page 5.

MONTGOMERY – Nowhere in the paper is any principle preceded by the term “Great” or “Important”.  These are adjectives that the reviewer applied in a pejorative and sarcastic manner. Is this appropriate for any scientific journal?  While the reviewer’s distain for great thinkers in history is his or her prerogative, this is very narrow minded, counterproductive, and arguably, solipsistic.    Some of the greatest neuroscientists wrote of great admiration of the philosophers.  Neuroscientists such as Sir Charles Sherrington knew and cited philosophers such as Descartes, demonstrating his appreciation and more importantly, the relevance of these philosophers to neuroscience.  The great physiologist, Claude Bernard, said to the effect that we are more often fooled by what we think we know than that which we do not.  The great value of philosophy, such as Aristotle, is that they challenge us to re-examine our presuppositions and assumptions.  The article attempts to do the same for the issues at hand.  We are not sure that the reviewer appreciates the significance of Kuhn’s notion of “paradigm” shift as discussed in The Structure of Scientific Revolutions.  It is far more than just a change in the prevailing perspective.  Moreover, scientists throughout the world over appreciate the importance of concepts such as “paradigm shifts” as evidenced by the praise and celebration given by Science and Nature on the 50th anniversary of Kuhn’s work.  Some of this article is written following Kuhn’s great work, that is an historical analysis of the ideas related to the notion of frequency effects in DBS.  Following from Santayana, I say if you want to change the future (in this case meaning greater understanding of DBS frequency) then you must see the future and the best way to see the future is to clearly see the past.

Reviewer 1 -   If this is meant to be a critical review of the literature in this topic (see prior comment), primary data other than that of the authors' own papers should also be considered and presented.

MONTGOEMRY – The reviewer fails to appreciate the difference between a critique and a review.  The primary purpose was to demonstrate that the problems with current understanding regarding the effects of DBS frequency in large degree related to conceptual errors and that future successful understanding is not likely if these conceptual errors are allowed to persist.  We describe our alternative possible theory, not so much to argue its validity but to serve as a contrast to highlight the issues.  As our theory is novel, there is a paucity of literature, other than our own, that bears on the alternative theory described.

Reviewer 1 -   The theory around line 332, that there is reason to suspect a specific interaction of an intrinsic motor circuit frequency and the preferred clinical DBS frequency, should really be developed much earlier in the article, and should be expounded upon further here.

MONTGOEMRY – we disagree as stated in our responses above.

Reviewer 1 -   The authors make a comment early on that a limited gridding of frequency leads to an under-appreciation of the effects of frequency. They then proceed to multiple examples where a handful of frequencies are used, generally spaced by tens of Hz. Figure 4 specifically labels a category of DBS as "high" without breaking it down further by granular frequencies, precisely what the authors claim the field has been doing wrong all these years. Where is the explanation for this apparent inconsistency?

MONTGOEMRY –  We clarified the description of what is high frequency in the revised text.  However, the reviewer is conflating his criticisms inappropriately resulting in the reviewer’s confusion.

Reviewer 1 -   There really needs to be some kind of table summarizing how the cited evidence base stacks up against various structures. The authors jump around to different basal ganglia structures in each example, making it hard to believe that they are finding general principles.

MONTGOMERY – this criticism is incorrect.  Again, the intent as re-emphasized in the revision is a critique of fundamental experimental and conceptual issues and particularly, to provide contrasting observations and alternatives.  It is not meant as an extensive review.  Further, the intent was to foster new and novel future studies that will clarify the issues.  Thus, it is not surprising that there is no listing of general principles, given the confused state of knowledge related to DBS frequency.  Any such attempt would be premature and counterproductive.

Reviewer 1 -   It is not clear how the primary data of Figure 15 were corrected for multiple statistical comparisons. (It is also not clear how those data support the idea of the reentrant loop.)

MONTGOMERY – the cut off of a z-score of 1.96 (two standard deviations) is both sufficiently robust and conservative to sustain multiple comparisons.  Certainly, the reviewer must be aware of the problematic nature of simple corrections for multiple comparisons resulting in a high risk of type II errors.

Reviewer 1 -    The captions/legends of Figures 16-17 are not at all clear to me. What is supposed to be different here? Why should randomizing the times of a spike train lead to a change in the ISI distribution? Are these subplots individual neurons, which is what the labels imply?

MONTGOEMRY – the explanation is contained within the text which is adaquete.

Reviewer 1 -   The article ends rather abruptly after a chunk of primary data, without ultimately summarizing what the authors' theory is and what they think should be done next.

MONTGOMERY – see comment above.

Reviewer 1 -   There are typos, e.g. "widow" for "window", that should be ground out by further proofreading.

MONTGOMERY – we corrected the noted typo. 

Reviewer 1 -   Broadly, at the end of reading this, I am left with a sense that I have been given a tour of the senior author's major papers, but with no clear unifying theory other than "things in the motor system oscillate, so DBS probably interacts with them". That is a perfectly valid theory, but it is not a paradigm shift, and it is not particularly falsifiable when given at this level of generality. This article needs to be re-written to clearly identify what has been done beyond the authors' own work, state their idea about a new model, then present the core components of that model.

 MONTGOMERY – That the reviewer “take away’ is so limited has more to do with the reviewer’s unwarranted antagonism and a perspective and expectation that is inappropriate.  When the situation is confused and unclear, it would be inappropriate to suggest that is it is not so.  The purpose was to provide a critique that illustrates why the situation is so confused and the authors’ model is only to serve to contrast current theories.  Paraphrasing Bertrand Russell, “If one cannot stand uncertainly one becomes religious, if one can stand uncertainly one becomes philosophical (and I would add, scientific).”

Reviewer 2 Report

Thanks for asking me to review this manuscript.

The authors present their hypotheses regarding the relationship of DBS effect, DBS frequency and oscillatory activity in the basal ganglia cortical circuits. The text is well written and structured although somewhat lengthy. The main message being that DBS acts as a “noisy” oscillator  signal that prevents pathological oscillations around a resonating circuit..... I have some comments to try and improve the manuscript.

The authors should acknowledge and try to encompass more of the clinical observations in DBS patients. In clinical practice, therapeutic effects are not always linked so clearly  with DBS frequency. Beneficial effects of STN DBS on e.g. rigidity can be seen with every frequency from 60Hz  to 130Hz+ . While there may be peaks at which clinical effects are optimal, above a certain frequency there do not appear to be frequencies associated with deleterious effects.

An increasingly important clinical observation supported by an increasing number of publications, is that lower frequencies eg 60-80Hz can have beneficial effects in comparison to 130Hz stimulation on speech and gait. This should be discussed.

Section 9.

I found this section, the least “reader-friendly”. To help make the text more accessible, this section could benefit from perhaps a couple of  additional sentences to more clearly explain what was done and why eg the purpose of the comparison between "antidromic-only" and "randomised-antidromic only" .

Typographical errors.

Line 329- Oralis

Line 415- Window

Line 607- neurons

Author Response

Reviewer 2

Reviewer 2 -   The authors present their hypotheses regarding the relationship of DBS effect, DBS frequency and oscillatory activity in the basal ganglia cortical circuits. The text is well written and structured although somewhat lengthy. The main message being that DBS acts as a “noisy” oscillator  signal that prevents pathological oscillations around a resonating circuit..... I have some comments to try and improve the manuscript.

Reviewer 2 -   The authors should acknowledge and try to encompass more of the clinical observations in DBS patients. In clinical practice, therapeutic effects are not always linked so clearly  with DBS frequency. Beneficial effects of STN DBS on e.g. rigidity can be seen with every frequency from 60Hz  to 130Hz+ . While there may be peaks at which clinical effects are optimal, above a certain frequency there do not appear to be frequencies associated with deleterious effects.

MONTGOMERY – We will make the corresponding changes in the paper.  Unfortunately, the paper is already quite long so we must be circumspect in responding to the suggestion.  The important point at the heart of the reviewer’s comment is that the clinical responses to DBS are complex and with the revisions, this has been demonstrated.  Again, this paper is not intended to be a comprehensive review of the past but a critique to help future efforts.

Reviewer 2 -   An increasingly important clinical observation supported by an increasing number of publications, is that lower frequencies eg 60-80Hz can have beneficial effects in comparison to 130Hz stimulation on speech and gait. This should be discussed.

MONTGOMERY – see comment above

Reviewer 2 -   Section 9.

I found this section, the least “reader-friendly”. To help make the text more accessible, this section could benefit from perhaps a couple of  additional sentences to more clearly explain what was done and why eg the purpose of the comparison between "antidromic-only" and "randomised-antidromic only" .

MONTGOMERY – we greatly appreciate the comment and have added additional clarifications.

Reviewer 2 -   Typographical errors.

MONTGOMERY – thanks for identifying, they have been corrected

Line 329- Oralis

Line 415- Window

Line 607- neurons